# Assessing RNA-Seq Workflow Methodologies Using Shannon Entropy

**DOI:** 10.3390/biology13070482

**Published:** 2024-06-28

**Authors:** Nicolas Carels

**Affiliations:** Laboratory of Biological System Modeling, Center of Technological Development in Health (CDTS), Oswaldo Cruz Foundation (Fiocruz), Rio de Janeiro 21040-900, RJ, Brazil; nicolas.carels@fiocruz.br; Tel.: +55-21-2598-4242

**Keywords:** RPKM, median normalization, benchmarking, entropy, PPI network, cancer, 5-year OS

## Abstract

**Simple Summary:**

We show how the relationship between the sub-network entropy of malignant up-regulated genes in twelve different types of cancer, spanning the entire spectrum of 5-year overall survival rates, can serve as a benchmark for optimizing RNA-seq workflows. Assessing the Shannon entropy of sub-networks formed by malignant up-regulated genes by several RNA-seq workflow approaches, such as DESeq2 and edgeR, but also by evaluating nine normalization methods on paired samples of TCGA RNA-seq, we found that the pipeline incorporating TPM normalization coupled with log_2_ fold change yielded the best correlation coefficient between cancer aggressiveness and tumor entropy.

**Abstract:**

RNA-seq faces persistent challenges due to the ongoing, expanding array of data processing workflows, none of which have yet achieved standardization to date. It is imperative to determine which method most effectively preserves biological facts. Here, we used Shannon entropy as a tool for depicting the biological status of a system. Thus, we assessed the measurement of Shannon entropy by several RNA-seq workflow approaches, such as DESeq2 and edgeR, but also by combining nine normalization methods with log_2_ fold change on paired samples of TCGA RNA-seq representing datasets of 515 patients and spanning 12 different cancer types with 5-year overall survival rates ranging from 20% to 98%. Our analysis revealed that TPM, RLE, and TMM normalization, coupled with a threshold of log_2_ fold change ≥1, for identifying differentially expressed genes, yielded the best results. We propose that Shannon entropy can serve as an objective metric for refining the optimization of RNA-seq workflows and mRNA sequencing technologies.

## 1. Introduction

The utilization of RNA sequencing (RNA-seq) has advanced significantly in cancer research and therapy over recent years [1,2]. RNA-seq, entailing thorough sequencing of RNA transcripts, was initially introduced in 2008 [3,4]. The primary objectives of RNA-seq analyses include the identification of differentially expressed and co-regulated genes, along with the inference of biological significance for subsequent investigations. Bulk RNA-seq employs a tissue or cell population as its starting material, yielding a blend of distinct gene expression profiles from the subject material under study. The transcriptomic landscapes of tumors exhibit considerable heterogeneity both among tumor cells, attributable to somatic genetic modifications, and within tumor microenvironments, arising from substantial stromal infiltration and the presence of diverse cell types within the tumor [5].

The domain of RNA-seq encounters persistent challenges, particularly concerning data processing and analysis. Unlike the microarray domain, which has seen a convergence of data processing methodologies into well-defined, widely accepted workflows over time, RNA-seq presents a continuously expanding array of data processing workflows, none of which has yet achieved standardization [6,7]. This situation partly arises from the diverse applications of RNA-seq, which may deviate from the underlying assumptions of the analytical methods employed [8], as real-world data often exhibit variations beyond those accommodated by theoretical models. Additionally, the verification of theoretical distributional assumptions remains challenging and can engender controversy [9]. Consequently, only a limited number of such signature panels have successfully transitioned into clinical practice due to issues of reproducibility. Nonetheless, it is recognized that certain genes exhibit consistent expression patterns within tumors [10,11], despite substantial intra- and inter-variability. These genes hold better promise for improved prognostics [5].

The primary method for assessing normalization techniques involves comparing the outcomes of raw and normalized data with quantitative real-time PCR (qRT-PCR), widely regarded as the gold standard for determining true expression values [12]. Although qRT-PCR has long served as a reference in numerous investigations, it is not flawless as an expression measurement assay itself, making it uncertain a priori which technology currently yields the most precise expression estimates [13]. In a considerable portion of the methods examined, genes exhibiting inconsistent expression across independent datasets tended to be smaller, possess fewer exons, and exhibit lower expression levels compared to genes with consistent expression measurements [6]. However, when evaluating relative quantification performance, several workflows displayed high expression correlations between RNA-seq and qRT-PCR expression intensities, indicating a generally high level of concordance between RNA-seq and qRT-PCR, with nearly identical performance observed across individual workflows [6]. Given that weakly expressed genes are typically utilized as a reference by parametric methods for normalizing RNA-seq data across samples, it is unsurprising that the detection sensitivity for their differential expression is relatively low. In this regard, non-parametric methods exhibit superior performance but may be susceptible to outliers with high expression levels [14]. However, depending on sequencing coverage, genes showing high levels of differential expression are more likely to be detected, leading to a convergence of results across methods [15]. Additionally, it appears that short reads facilitate simpler methodological workflows compared to longer reads [16,17].

To address biological and methodological variations within a user-friendly framework, an expanding array of open-access semiautomated pipelines is emerging online. Examples include iDEP [18], LVBRS [19], RNAseqChef [20], and NormSeq [21]. It has been observed that achieving consensus among pipelines enhances the diagnostic accuracy of differentially expressed genes (DEGs), suggesting that combining diverse methodologies can yield more robust results [22].

The objective of normalization is to mitigate or remove technical variability. A prevalent approach, shared among numerous normalization methods, involves redistributing signal intensities across all samples to ensure they exhibit identical distributions [23]. An essential step in an RNA-seq analysis is normalization, where raw data are adjusted to account for factors such as total mapped reads and coding sequence (CDS) size. Errors in normalization can greatly impact downstream analyses, leading to inflated false positives in differential expression studies [8]. The distortions produced include false effects (false positives), effect-size reduction, and masking of true effects (false negatives), as demonstrated by Wang et al. [24]. For instance, raw counts are often not directly comparable within and between samples [14]. Additionally, other stages of RNA-seq processing throughout the pipeline execution may also impact outcomes. A recurring challenge is assessing the reliability of RNA-seq processing and the confidence level associated with downstream findings. An essential consideration in the comparison of normalization methods is to ascertain which method most effectively retains biological veracity [12]. While several normalization methods [25] and processing techniques [26] have been compared, discrepancies between them remain unclear.

Another approach that has been pursued is normalization by referencing housekeeping or spike-in genes [21]. Housekeeping genes are assumed to exhibit consistent expression levels across samples from diverse tissues, and it has been demonstrated that normalizing qRT-PCR data using conventional reference genes yields comparable results to those obtained using stable reference genes selected from RNA-seq data [27]. However, the notably small dispersions and proportion of DEGs in spike-in data could yield substantially varied benchmarking results [28], rendering this technique unreliable [29].

There are essentially two categories of RNA-seq normalization methods [14]: (i) non-parametric methods that do not impose a rigid model of gene expression to be fitted. These methods implicitly consider that data distribution cannot be defined by a finite set of parameters, so the amount of information about the data can increase with its volume [22]. An example of non-parametric methods is Reads per Kilobase Per Million Mapped Reads (RPKM, [30]), where counts of mapped reads are normalized by reference to the total read number and CDS size. These methods also include RPKM variations such as FPKM and TPM [12,31], and (ii) parametric methods entail the mapping of expression values for a specific gene into a specific distribution, such as a Poisson or negative binomial. One example of a parametric method is DESeq2 [32], which normalizes count data and estimates variance using a negative binomial distribution model [33]. This approach is predicated on the assumption that the majority of genes are not differentially expressed, and it accommodates variations in sequencing depth across samples.

Parametric and non-parametric methods can be used to assess the differential expression on a *gene-by-gene* basis or on a *population-wide* basis. Following normalization, differential expression analysis on a *gene-by-gene* basis is conducted by log transformation to ascertain fold changes, expressed as positive (up-regulation) or negative values (down-regulation). The classification threshold for fold changes is determined according to the logarithm base of 2. Therefore, a log fold change of 1 corresponds to a twofold difference in expression, while a log fold change of 2 corresponds to a fourfold difference in expression, etc.

In the *population-wide* approach, as utilized by Carels et al. [34], the threshold for differential expression is established by referencing the overall population of DEGs, which is modeled by fitting a Gaussian function to the observed distribution of DEGs. According to this methodology, a gene is categorized as being up-regulated (or down-regulated) if its normalized raw count exceeds a critical value determined based on a user-defined *p*-value. Consequently, for a gene to be classified as up-regulated, its level of differential expression must surpass that of the majority of other genes within the population (*population-wide*).

The method of evaluating the expression of genes by reference to the population of DEGs has been used to identify up-regulated hub targets in solid tumors [10,11,34,35,36].

Through the utilization of this approach, Conforte et al. [10] reaffirmed the correlation observed by Breitkreutz et al. [37] between the entropy degree of protein–protein interactions (PPI) and cancer aggressiveness, initially discovered using KEGG, this time employing RNA-seq data sourced from TCGA.

Here, we propose utilizing the negative correlation between the entropy of the PPI sub-network of malignant up-regulated genes and tumor aggressiveness, quantified by the 5-year overall survival (OS) rate of patients, as a benchmark for evaluating RNA-seq processing methods. We validated this process in three steps: (i) We evaluated the performance of RPKM and median normalization on a *gene-by-gene* basis and by referencing the population of DEGs across eight types of cancer (475 patients), spanning 5-year OS rates from 30% to 98%, according to previous studies [10,11]. (ii) We compared RPKM to seven read count normalizations, i.e., transcript per million (TPM, [38]), counts per million (CPM, [14]), median (Med, [14]), upper quantile (UQ, [39]), relative log expression (RLE, [32]), quantile normalization (QN, [40]), and trimmed mean of M-values (TMM, [41]), as well as to two cross-sample distribution-based methods, i.e., DESeq2 [32] and edgeR [42]. (iii) Based on the best performing methodologies identified in these comparisons, we tested paired samples from additional cancer types from TCGA to validate the approach, including bladder carcinoma (BLCA), lung adenocarcinoma (LUAD), colon adenocarcinoma (COAD), and uterine carcinoma (UCS).

According to this Bayesian learning process, we found the following: (i) The coefficient of correlation between average entropy per cancer type and aggressiveness (5-year OS) is a suitable metric for the comparative performance of biological information extraction from sub-networks of up-regulated malignant genes. (ii) TMM, QN, and RLE, a group of methods that determine a scaling factor for variation stabilization, produced a correlation coefficient similar to TPM but with a standard deviation approximately 25% lower. The straightforward approach of combining a normalization method (even without a scaling factor for variation stabilization) with log_2_ fold change yielded better average correlation coefficients than probabilistic methods such as DESeq2 and edgeR for determining network entropy. (iii) The correlation coefficient decreased from *r* ≈ −0.94 to *r* ≈ −0.67 when the number of cancer types increased from 8 to 12.

## 2. Materials and Methods

### 2.1. RNA-Seq

The gene expression data were acquired in the form of RNA-seq files (raw counts) of paired samples (malignant and healthy tissue from the same patient from a cohort of 515 individuals; Appendix A) from the GDC Data Portal (https://portal.gdc.cancer.gov/, accessed on 1 March 2020, see [11]). The data sourced from GDC are presented in Table 1 and account for 12 different types of cancer.

RNA-seq profiles were available for 60,483 GDC sequences (Ensembl accessions). However, to compute entropy, we needed PPIs that were extracted from the 2017 version of IntAct (https://www.ebi.ac.uk/intact/download/ftp, accessed on 11 January 2018). Given that IntAct PPIs are given in UniprotKB accessions, the process of establishing equivalence between Ensembl and UniProtKB accessions (Esembl2UK step) was limited to 15,526 genes (~75% of the human proteome). Consequently, it was this latter dataset that underwent the entire comparative analysis.

### 2.2. Overall Survival

The 5-year survival rates of the malignant tissues were inferred based on the overall survival (OS) data available from the Cancer Genome Atlas Clinical Data Resource (TCGA-CDR) [43], which contains curative clinical and survival data from TCGA patients specifically designed to eliminate incomplete survival (follow-up) information. Appendix A of Liu et al. [43] has two columns, “OS” and “OS.time”, that were used in GraphPad Prism (Boston, MA, USA) software version 5.02 for survival curve analysis for BLCA, COAD, and UCS, indicating death/event as 1 and censored data as 0. This analysis resulted in survival rates corresponding to days to “death/last follow-up” for each cancer type (Appendix A). The survival rate observed over 5 years (1200 days) was used to represent each cancer type. The 5-year OS of STAD, LUSC, LUAD, LIHC, KIRC, KIRP, BRCA, THCA, and PRAD were drawn from Table S4 (https://www.frontiersin.org/articles/10.3389/fgene.2019.00930/full#supplementary-material, accessed on 10 May 2024) of Conforte et al. [10].

### 2.3. Basic Normalization Methods

**RPKM**: The gene expression in reads per kilobase (*RPK*) of a gene is defined as the ratio of the number of reads mapped to it over its length (in kilobases) and *RPKM* as the ratio of the *RPK* of a gene over the total reads in the sample (in millions) [30].

Here, we computed *RPKM* (*RPKM* normalization step) according to Formula (1)
(1)$$RPKM=\frac{{RC}_{g}}{L\times \left({RC}_{pc}-\left(\delta \times {RC}_{pc}\right)\right)}\times 10^{9}$$
where:

*RCg*: number of reads mapped to the gene;

*RCpc*: number of reads mapped to all protein-coding genes;

*L*: size of the coding sequence in base pairs;

*δ*: A tuning factor such that when *δ* = 0, Formula (1) is equivalent to the standard RPKM. In this work, we used *δ* = 0.95 because it optimized the coefficient of correlation between entropy and 5-year OS.

**TPM**: The gene expression in transcripts per million (TPM) of a gene is defined as the ratio of its *RPK* over the *sum of all RPKs* (*per million*) [38]. We computed raw counts in accordance with Formula (2) using a custom Perl script derived from the one used for RPKM calculation.
(2)$${TPM}_{i}=\frac{{RPK}_{i}}{\sum_{i=1}^{n}{RPK}_{i}}\;{\times 10}^{6}$$

**Median**: We used a custom Perl script to implement the procedure described in https://scienceparkstudygroup.github.io/rna-seq-lesson/05-descriptive-plots/index.html#43-deseq2-normalized-counts-median-of-ratios-method (accessed on 10 May 2024) for normalizing paired samples of raw counts obtained from GDC RNA-seq data (median normalization step).

### 2.4. Extended Normalization Methods

In this section, we aim to compare the normalization methods provided by the NormSeq server (https://arn.ugr.es/normSeq, accessed on 10 May 2024) to RPKM and TPM. Briefly and citing Scheepbouwer et al. [21], the purpose of these methods can be summarized as follows:

**Counts Per Million (CPM):** CPM normalization corrects for library size without considering transcript length [14].

**Median (Med)**: median normalization adjusts the data of each individual sample by adding a constant value to achieve the same median value across all samples [14].

**Upper Quantile (UQ)**: all genes with a read count of 0 are removed, followed by a division of the remaining gene counts by the upper quartile [39].

**Relative log expression (RLE)**: for each gene, the RLE scaling factor is computed as the median of the ratio of the read counts by taking the geometric mean across all samples [32].

**Quantile normalization (QN)**: quantile normalization applies a mathematical transformation to the rank statistics across samples [40].

**Trimmed mean of M-values (TMM)**: the TMM method estimates scale factors for comparing libraries on a relative scale [41].

### 2.5. Differential Expression Method

Here, we considered DESeq2 [32] and edgeR [42] as reference software for benchmarking the capacity of entropy to report on the extraction of biological information given the complexity of sub-networks associated with malignant up-regulated genes. Both packages are cross-sample distribution-based methods that estimate the dispersion parameter for each gene, reflecting the variability of read counts according to the negative binomial distribution. This software applies the Benjamani–Hochberg procedure to control the false discovery rate (FDR), helping manage the multiple testing problems inherent in RNA-seq data analysis (cf. https://bioconductor.org/packages/release/bioc/manuals/DESeq2/man/DESeq2.pdf, accessed on 10 May 2024). However, they differ in their normalization methods: DESeq2 is based on the median of ratios to normalize read counts (MRN), and EdgeR is based on the trimmed mean of M-values (TMM). DESeq2 was run from the iDEP server (http://bioinformatics.sdstate.edu/idep/, accessed on 10 May 2024) [18] and edgeR from NormSeq (https://arn.ugr.es/normSeq, accessed on 10 May 2024).

### 2.6. Up-Regulated Genes

**Gene-by-gene**: Log_2_ fold change was computed with a custom Perl script (log fold change step). Genes exhibiting a differential expression exceeding log_2_ > +1 (fold change = 2) were categorized as up-regulated.

**Population-wide**: To identify significant DEGs in the tumor samples, we subtracted the gene expression values of control samples from their respective tumor-paired samples. The resulting values were referred to as differential gene expression (DEG step). Negative differential gene expression values indicated higher gene expressions in control samples, while positive differential gene expression values indicated higher gene expressions in tumor samples.

To expand the distribution of DEGs, we eventually applied a log transformation (xlogx step).

The Gaussian function was fitted onto the normalized differential expression with the Python package scipy. Probability density and cumulative distribution functions (PDF and CDF, respectively) were computed within the range of differential gene expression from −30,000 to +30,000 to calculate the critical value (CVC step) corresponding to a one-tail cumulative probability *p* = 0.975, equivalent to a *p*-value α = 0.025. Genes were categorized as up-regulated if their differential expression exceeded the critical value associated with *p* = 0.975. The range of −30,000 to +30,000 was suitable for the *p*-value and normalization conditions outlined in this report.

In a subsequent step, the protein–protein interaction (PPI) sub-networks were inferred for the proteins identified as products of up-regulated genes (obtained by *gene-by-gene* or population approaches). The sub-networks were derived by cross-referencing these gene lists with the human interactome (SRC step).

The human interactome (151,631 interactions among 15,526 human proteins with UniProtKB accessions) was obtained from the intact-micluster.txt file (version updated December 2017), accessed on 11 January 2018.

We used the PPI sub-networks of up-regulated genes from each patient to determine the connectivity degree of each vertex (protein) by automatically counting their edges (CC step). These metrics were used to compute the Shannon entropy (ETP step) of each PPI sub-network, as elaborated in the section entitled “Shannon Entropy” below.

### 2.7. Shannon Entropy

Shannon entropy is a suitable measure to calculate the complexity or information content of networks (see Zenil et al. [44]); it was calculated with Formula (3)
(3)$$H=-\sum_{k=1}^{n}p\left(k\right){log}_{2}(p\left(k\right))$$
where *p*(*k*) is the probability of occurrence of a vertex with a rank order *k* (*k* edges) in the sub-network considered. The sub-networks were generated automatically from gene lists found to be up-regulated in each patient. The edges between sub-network vertices were established by referencing the interactions described in the IntAct interactome [45]. Subsequently, the sub-network vertices were listed and counted. The events of *k* edges were computed from the minimum value of one edge between two vertices to the event corresponding to the vertex with the maximal edges *n* and its neighbors for the sub-network under consideration. When *k* did not match any vertex in the network, its corresponding entropy was not computed because it would result in an undefined value (log_2_(0) is undefined). All the Perl scripts involved in this process were custom-made and described in Pires et al. [11]. The script for entropy calculation produced results identical to those obtained using the *Entropy* function of R (https://www.rdocumentation.org/packages/DescTools/versions/0.99.54/topics/Entropy, accessed on 10 May 2024).

### 2.8. Statistics

The correlations were obtained with the classical formula r = cov(X,Y)/σXσY and orthogonal regression lines as reported by Jolicoeur [46]. The scripts of this report can be downloaded from GitHub: https://github.com/BiologicalSystemModeling/Theranostics, accessed on 10 May 2024 under the MIT License.

## 3. Results

### 3.1. Step 1: Assessment of Normalization on a Gene-by-Gene or Population-Wide Basis

When comparing the up-regulated genes as computed by the gene-by-gene approach for RPKM (Figure 1A) and median (Figure 1B) normalization methods, we obtained the plots of Figure 1C and Figure 1D, respectively.

When considering log fold change, the correlation coefficient improved with RPKM normalization (*r* = −0.91; Figure 1C) compared to the Mednorm approach (*r* = −0.80; Figure 1D) in the gene-by-gene analysis. Although the slope associated with the pipeline in Figure 1C is greater than that of Figure 1D, the disparities in correlation between both relationships are not striking.

When comparing the correlation coefficients of both normalization methods within the population-wide approach (Figure 2A,B), it is evident that the linear regression associated with the negative correlation by RPKM (Figure 2C) is maintained, albeit slightly lower (*r* = −0.84).

Conversely, in Figure 3D, the linear regression linked to the negative correlation by Mednorm is lost as the correlation coefficient does not exceed *r* = −0.16, indicating a loss of discrimination power for highly expressed genes. This loss of discrimination power for highly expressed genes can be attributed to the use of the median to mitigate the variance introduced by these genes.

The effect of the xlogx transformation is depicted in Figure 4, where the histogram of the RPKM pipeline without xlogx transformation is shown in Figure 4A, and the same pipeline with the inclusion of the xlogx step is presented in Figure 4B. The addition of the xlogx step results in a flattening and broadening of the DEG distribution, enhancing the list of genes categorized as up-regulated.

The flattening and broadening of the DEG distribution is correlated with that of the δ tuning factor. The native RPKM formula (δ = 0) produces a very narrow distribution, while with δ = 0.95, the normalized counts are inflated, which results in spreading the DEG distribution for the pipelines of Figure 2A or Figure 3A. Thus, as δ increases above 0, the DEG distribution becomes narrower, the critical value associated with the same *p*-value decreases, the size of the up-regulated gene list decreases, and the entropy decreases. The entropy reduction is expected from the fact that the probability of drawing a hub with a high connection degree is lower in a small list than in a large one. By contrast, varying δ had no effect on the size of the up-regulated gene list in the Figure 1A pipeline since, whatever the normalized value of read counts, the proportion between the malignant and reference RNA-seq through log_2_ fold change remains the same. Thus, tuning δ was only effective for the *population-wide* approach but not for the *gene-by-gene* one.

To gain a deeper insight into the impact of computing differential expression on a *population-wide* basis versus a *gene-by-gene* approach, let us consider the practical scenario of BRCA: (i) A gene may exhibit expression levels in tumors that are at least two times higher (fold change ≥ 2) compared to its corresponding normal tissue, where its expression level may be close to zero. Despite being expressed at least two times higher in tumors compared to normal tissue, it may still be expressed at a low level if compared to other DEGs after normalization. This example illustrates that such a gene might not be categorized as up-regulated by a *population-wide* approach, whereas it would be by a *gene-by-gene* approach. An instance of this case is MKI67, whose normalized expression levels were <x> = 1541.9 (σ = 1034.7) in the tumor and <x> = 261.4 (σ = 190.3) in the normal tissue. (ii) The *population-wide* approach considers DEGs to be significant when the expression difference between the tumor and the normal tissue surpasses that of the DEG population based on a specified *p*-value threshold. In other words, the absolute value of the expression difference in the tumor does not necessarily need to be at least two times greater than that in the normal tissue; it simply needs to exceed the critical value associated with the chosen *p*-value. An example of this is the chaperone HSP90AB1, whose normalized expression levels were <x> = 16,748.9 (σ = 5174.2) in the tumor and <x> = 12,491.7 (σ = 2865.3) in the normal tissue. (iii) Considering the xlogx transformation, the resulting distribution flattening and broadening amplifies (by a factor of 3) the critical value associated with a specific *p*-value. This alteration may influence its association with the classification of a gene as up-regulated or not, as indicated by the higher entropy observed when compared to pipelines lacking the xlogx step. For instance, considering a *p*-value of 0.025, the critical values were <x> = 3104.9 (σ = 371.7) for the RPKM pipeline without the xlogx step and <x> = 10,900.7 (σ = 1364.2) for that pipeline including it.

### 3.2. Step 2: Comparison of Normalization Methods and Differential Expression Determination Processes

The average entropies in Table 2 were calculated from each paired sample (Appendix A) by replacing RPKM by TPM in the pipeline of Figure 1A and replacing the normalization method step in Figure 1B with UQ, Med, CPM, RLE, QN, or TMM.

Table 2 shows that the correlation coefficient between entropy and 5-year OS for the sub-networks of malignant up-regulated genes of eight cancer types was lower for DESeq2 and edgeR (*r* = −0.72) compared to the correlation obtained with a log_2_ fold change ≥ 1 filter applied to the raw count normalized with any method (even with no normalization). RLE and PCA plots are given in Appendix A for DESeq2.

This table also shows that TPM (*r* = −0.94) ranks highest among methods without variance stabilization by a scaling factor, such as RPKM (*r* = −0.91), UQ (*r* = −0.81), Med (*r* = −0.93), and CPM (*r* = −0.77). Some of these methods performed even better than variance-stabilized methods (RLE: *r* = −0.91, QN: *r* = −0.90, TMM: *r* = −0.91), but they exhibited nearly double the rate of average standard deviation (0.4 to 0.6 compared to ~0.3).

The variance stabilization of RLE, QN, and TMM normalizations can be verified from their RLE plots [47] compared to UQ, Med, and CPM (Appendix A). However, variance stabilization does not necessarily improve the PCA classification of control and tumor samples (Appendix A); in some cases, the PCA classification is effective, and in others, it is not, without any apparent correlation to any specific feature.

### 3.3. Step 3: Generalization of the Degree-Entropy vs. 5-Year OS Relationship

Since the eight cancer types in Table 2 could be the result of an over-fitting process, we included paired samples of four additional cancer types: BLCA, LUAD, COAD, and UCS. Table 3 shows that it is indeed the case; however, the negative relationship between entropy and 5-year OS is maintained. The best result was obtained with TPM (*r* = −0.674), while RLE (*r* = −0.602) and TMMM (*r* = −0.598) exhibited lower correlation coefficients but were similar and had a lower variance.

By plotting the relationship entropy vs. 5-year OS for TPM (Figure 5A) and RLE (Figure 5B), one can better visualize the lower variation associated with RLE compared to TPM. Both relationships are very similar. The decrease in the correlation coefficient from TPM to RLE or TMM is due to the decrease in covariance. The covariance for TPM was −6.458, while for RLE, it was −5763. From this, one may conclude that variance stabilization through the application of a scaling factor has a negative effect on the correlation.

## 4. Discussion

### 4.1. Biological Significance of Degree-Entropy

As highlighted in Abrams et al. [12], the primary objective of RNA-seq processing should be the preservation of biological signals. However, given the intricate nature of biological systems, careful consideration must be given to minimize processing biases while accurately capturing these signals. While RNA-seq offers insights into gene expression, it captures only a fraction of biological characteristics and thus falls short of fully encompassing biological signals. It is therefore argued that a system approach to gene expression is necessary to achieve comprehensive understanding.

Linking RNA-seq data to the interactome provides a system-level dimension by integrating gene expression data with the topological structure of the biological system under investigation. The interactome facilitates a synthetic transformation of gene expression data into a representation of the biological system, serving as a benchmark. Nonetheless, questions arise regarding the common attributes of biological systems, and methodologies suitable for mathematically representing and evaluating RNA-seq processing performances.

Among the distinguishing features of biological systems, adaptability emerges prominently [48]. There exists a trade-off between adaptability and the robustness of biological networks [49], where robustness signifies stability and resilience denotes the ability to return to equilibrium after perturbation. Both attributes are crucial for the survival of biological entities [50]. Studies consistently demonstrated that resilience at lower organizational levels contributes significantly to the robustness of entire systems [51]. Key features of resilience at the network level encompass modularity, redundancy, and diversity [52], wherein (i) redundancy in pathways or isoforms enhances adaptation to varying environmental conditions, (ii) modularity represents dense clusters of connections between vertices, and (iii) diversity refers to the variety of elements within the system. These features can be depicted by network topology, specifically by the distribution of edges per vertex.

Hubs have been associated with protein essentiality based on their positional context within the network [53,54]. Additionally, hubs can act as bottlenecks, linking numerous inputs to a limited number of outputs. These bottleneck proteins, as pivotal connectors, are more likely to be essential [55]. The essentiality of central hubs within modules surpasses that of peripheral hubs [53,54], and inhibiting them exerts a more profound disarticulating effect on the network compared to less connected vertices [56,57]. In cancer contexts, hubs can rewire pathways through negative and positive feedback loops [50,51,58,59,60,61,62], leveraging their ability to interface with partners from diverse pathways [63]. Stochastic alterations in tumor cell transcriptional programs enhance their adaptative potential across varying conditions. Gurova [64] hypothesized that unstable chromatin facilitates stochastic transitions between transcriptional programs in aggressive cancers, potentially enabling the repurposing of existing signal transduction pathways [58,65]. Many cancer-related hubs function as chaperones, rectifying misfolded proteins arising from mutations, thereby decoupling genetic variations (mutations) from phenotypic expression and isolating low-level fluctuations from high-level functionalities (phenotype variation) [50]. These resilience mechanisms are needed for sustaining or even enhancing tumor aggressiveness within their environment [58,66].

These considerations underscore the importance of evaluating biological systems using networks, particularly PPI networks, in the context of cancer. Entropy serves as a metric for assessing the structural and dynamic properties of networks, which is biologically relevant because the macroscopic resilience of a steady state correlates with the uncertainty in the underlying microscopic processes, a property quantifiable through entropy [53]. Given that cancer-related hubs exhibit topological properties associated with the key biological functions discussed earlier, their evaluation within sub-networks of malignant up-regulated genes is crucial for elucidating tumor characteristics. The overexpression of hubs significantly contributes to an elevation in the overall entropy rate [67]. However, entropy alone lacks significance without contextualization. Therefore, accurately examining its correlation with patient 5-year OS, a measure of cancer aggressiveness, becomes paramount [10].

The biological systems represented by tumors across 12 cancer types, ranging in aggressiveness from 20% to 98%, exhibit considerable complexity and variability. Sources of variance in RNA-seq include (i) the tissue origin of samples, (ii) sequencing technology, (iii) sequencing equipment, (iv) sequencing personnel, (v) sequencing coverage, (vi) read size, and (vii) differential expression profiles. These factors collectively influence workflow performance, prompting questions about whether (i) the workflow can consistently achieve high success rates across samples of such diverse complexity or (ii) it can demonstrate superior performance with specific sample types. Ideally, a workflow capable of delivering reliable results across a broad spectrum of biological complexities is preferred. Maximizing the correlation coefficient between entropy and aggressiveness across this diversity of biological contexts is akin to identifying the optimal combination of technologies that performs effectively across the spectrum of variability.

It is well acknowledged that multiple genomic clonal populations within a neoplasm arise from divergent evolution of progeny cells, leading to increased tumor heterogeneity over time [68,69,70,71,72,73]. Significant temporal changes in transcriptome profiling and chromatin accessibility have been observed due to the emergence of distinct cell populations [74].

Given that tumor aggressiveness is known to correlate positively with complexity [75,76,77,78], entropy emerges as a suitable metric for benchmarking RNA-seq workflows. Here, we consider tumor heterogeneity and complexity comprehensively, focusing solely on bulk RNA-seq. In bulk RNA-seq, up-regulated genes reflect the average expression levels conserved across the various cell lineages within the tumor sample being sequenced [79]. If this sample is representative of the entire tumor, one may hypothesize that up-regulated genes identified in bulk RNA-seq represent the primary determinants of cancer—those that render the tumor compatible with its environment. Expanding on this, the quantitative concept of aggressiveness can be defined as a secondary determinant of biological systems, stemming from their ability to more or less effectively exploit an ecological niche [80]. In the context of cancer, this niche exists between the tumor and its host tissue. In contrast to genes whose overexpression is consistently significant throughout the tumor, those that are selectively overexpressed in specific cell lineages without a discernible impact on the overall tumor level could be regarded as secondary determinants.

As highlighted by Baltazar et al. [81], “mean entropies represent the average contribution from individual hubs”. Consequently, given that aggressive tumors exhibit heightened complexity and molecular heterogeneity, and considering that networks accurately portray tumor biology, entropy emerges as a pertinent measure of network topology (i.e., complexity) and aggressiveness, as demonstrated by Conforte et al. [10]. Furthermore, hubs should be acknowledged as key components influencing network entropy [56]. Therefore, we advocate for the importance of leveraging the relationship between tumor entropy and 5-year OS (or any other measure of cancer aggressiveness) as a benchmark for evaluating RNA-seq processing methodologies. Ultimately, it is imperative to identify the optimal workflow for diagnosing hubs that are most suitable for theranostic applications.

According to Hu et al. [82], hubs show significant enrichment in the PPI network of 18 classes of diseases, including those of the stomatognathic, endocrine, digestive, respiratory, female urogenital, nervous, and musculoskeletal systems, as well as cancers. Therefore, it is plausible to anticipate a correlation between network topology and the specific characteristics of these diseases. However, we would expect that a RNA-seq workflow optimized for cancer should also be applicable to RNA-seq from other biological systems. The main divergence lies in the read mapping process, primarily due to the frequent occurrence of mutation, fusion, and indel events in genes affected by malignant processes. Given that cancer poses the most demanding scenario for extracting biological information via RNA-seq, there is no apparent reason why a workflow tailored for tumor samples would not be suitable for RNA-seq aimed at characterizing other biological contexts.

### 4.2. Comparison of Gene-by-Gene and Population-Wide Approaches

The correlation observed between the entropy of up-regulated genes using the *gene-by-gene* approach and the 5-year OS (*r* = −0.91) with STAD, LUSC, LIHC, KIRC, KIRP, BRCA, THCA, and PRAD suggests that the PPI sub-network of up-regulated genes associated with aggressive cancer (LUSC, LIHC, and STAD) exhibits greater complexity. This complexity is characterized by an increased number of hubs and alternative pathways, providing higher redundancy when compared to less aggressive cancers (THCA, PRAD). The remarkably high correlation coefficient delineates three distinct groups of entropy versus aggressiveness, with KIRC, KIRP, and BRCA positioned in the middle. The heightened pathway redundancy observed in aggressive cancer serves as a mechanism for tumor resilience to therapeutics and a propensity for relapse. The negative correlation identified through the *gene-by-gene* approach corroborates findings from previous studies [10,37,83,84,85]. Notably, this correlation remains robust despite the various origins of the TCGA data, which were generated in different laboratories by disparate teams, using distinct sequencing technologies.

Moreover, the correlation level associated with the *gene-by-gene* approach underscores the relationship between the entropy of up-regulated genes and the 5-year OS, serving as an objective benchmark for refining bioinformatic pipelines and sequencing technologies. This benchmark aids in fine-tuning the pipelines to maximize the extraction of biological information from RNA-seq data with high precision, as detailed in this report.

The *gene-by-gene* approach extracts more up-regulated genes than the *population-wide* method because certain genes, which may be up-regulated by a factor of 2 in the tumor compared to control, might still exhibit low-level up-regulation on a *population-wide* scale. A filter for RPKM > 10 [7] did not change this picture significantly. In contrast, the difference in differential gene expression between the tumor and its paired reference is statistically greater than that obtained by the *gene-by-gene* approach.

We concluded from the above that the *population-wide* approach extracts fewer relevant genes in terms of up-regulation when compared to the *gene-by-gene* approach. However, the number of hubs taken into account by the *population-wide* approach is proportional to that of the *gene-by-gene* approach, which explains why the linear regression between entropy and 5-year OS remains consistent. The reason why the correlation coefficient is lower for the *population-wide* approach (*r* = −0.84) compared to the *gene-by-gene* approach (*r* = −0.91) is that the variance increases proportionally to the average gene expression. Since the average of gene expression is larger for the sample of up-regulated genes in the *population-wide* approach compared to the *gene-by-gene* one, it is expected that the correlation coefficient is lower for the *population-wide* approach than for the *gene-by-gene* one (given the average variance is larger); however, the linear regression is maintained.

When excluding the xlogx step from the RPKM pipeline, we observed a diminution of the correlation coefficient from *r* = −0.84 to *r* = −0.72. Since the diminution of the correlation coefficient is due to an increase in variance, we concluded the xlogx step has an effect on variance stabilization.

The median normalization produced a correlation with a lower correlation coefficient (*r* = −0.80) compared to RPKM, considering the *gene-by-gene* approach. However, when considering the *population-wide* approach, the median correlation disappeared (*r* = −0.16) even if the xlogx step was excluded from the pipeline (*r* = −0.17), which indicates that genes significantly up-regulated on a statistical basis are not suitably normalized by the Mednorm method. This suggests that a bias is introduced by this method of normalization in genes with extreme levels of gene expression.

### 4.3. Comparison of Normalization Methods and Differential Expression Determination Processes

By reference to entropy, we found that DESeq2 and edgeR were not effective in assessing the complexity of the biological network across a large cohort encompassing a 5-year OS aggressiveness rate ranging from 20 to 98%. From this finding, one might conclude that methods relying on negative binomial distribution and Benjamani–Hochberg procedure for false discovery rate (FDR) control are not optimal for extracting biological complexity from RNA-seq data. Conversely, normalization methods such as TPM, RLE, and TMM combined with log_2_ fold change appear suitable for this purpose.

### 4.4. The Relationship of Degree-Entropy and 5-Year OS

It is interesting to note the negative correlation between degree-entropy and 5-year OS, albeit this correlation diminishes notably when analyzing paired samples of 12 cancer types rather than eight. The underperformance of Med in this context indicates its inadequate adaptation to biological samples, characterized by significant topological complexity variation. This underscores the presence of overfitting in the previous analyses. The decrease in correlation observed for a larger set of cancer types suggests that tumors adjusted their gene up-regulation patterns to form more or less complex sub-networks depending on the cancer type and its environment. This consideration holds importance in guiding clinical decisions regarding optimal therapeutic strategies. For instance, therapies targeting hubs may be less effective in tumors exhibiting lower entropy levels.

Among the types of errors that could account for the correlation reduction across the 12 cancer types, it should be noted that the standard deviation associated with the *x* axis is unknown. Although 5-year OS is a statistical metric intended to ensure the robustness of the data on the *x* axis, uncertainties remain. Another factor to consider is that the efficacy of treatments for specific cancer types can vary independently of their aggressiveness. This variability can influence the 5-year OS, which has generally increased over time but at differing rates depending on the type of cancer.

## 5. Conclusions

In this report, we discuss the use of the negative correlation between the sub-network entropy of malignant up-regulated genes and 5-year OS as a benchmark to assess the efficiency of a workflow to extract information from raw-read counts. We believe the exercise is relevant because this negative correlation is a biological observation based on a cohort of 515 patients across 12 different cancer types that cumulates a variability that was not corrected. This exercise is interesting in the sense that it compares workflows covering different strategies and involving parametric and non-parametric normalization methods. We found that the pipeline incorporating TPM and RLE or TMM normalizations coupled with log_2_ fold change yielded the best correlation coefficient between cancer aggressiveness and tumor entropy. We also observed that the discrimination power of median normalization vanished for genes with high expression levels. The workflow configuration had a strong impact on the sub-network entropy of malignant up-regulated genes, consistent with biological observation. Here, we did not pretend to be exhaustive in method comparison, but rather to draw the readers’ attention to the potential of using this correlation to fine tune alternative workflows described in the literature.

## Figures and Tables

**Figure 1 biology-13-00482-f001:**
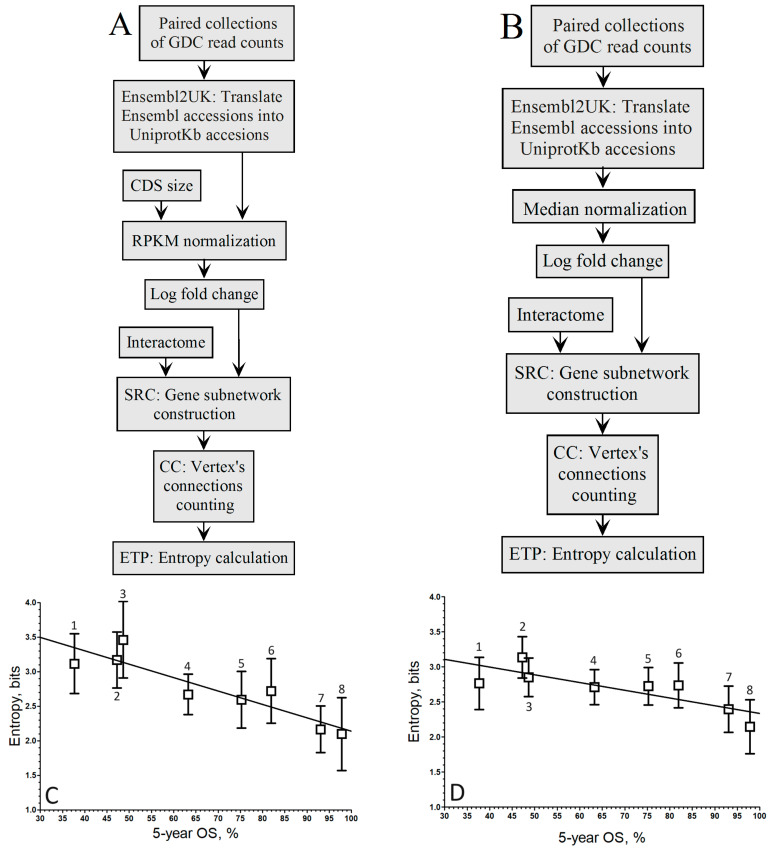
Pipelines to compute the relationship between the entropy of up-regulated gene networks in GDC-paired samples from the RNA-seq of 8 cancer types and their 5-year OS by the gene-by-gene approach. (**A**). Pipeline of RPKM normalization and log fold change. (**B**). Median normalization and log fold change. (**C**). RPKM (A pipeline; *r* = −0.91; y = −0.0196 x + 4.08). (**D**). Mednorm (B pipeline; *r* = −0.80; y = −0.0107 x + 3.41). ^1^ STAD, ^2^ LUSC, ^3^ LIHC, ^4^ KIRC, ^5^ KIRP, ^6^ BRCA, ^7^ THCA, ^8^ PRAD. The boxes represent the average entropy per cancer type, and the whiskers correspond to their standard deviations.

**Figure 2 biology-13-00482-f002:**
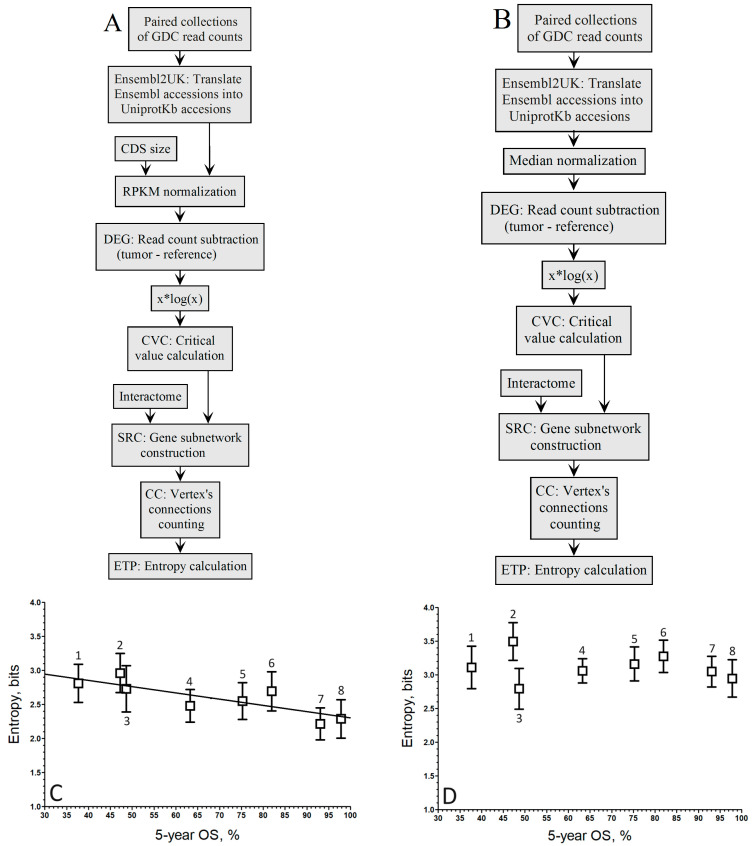
Pipelines to compute the relationship between the entropy of up-regulated gene sub-networks in GDC-paired samples from the RNA-seq of 8 cancer types and their 5-year OS by the population-wide approach. (**A**). Pipeline of RPKM normalization with the xlogx step. (**B**). Pipeline of median normalization with the xlogx step. (**C**). RPKM (A pipeline; *r* = −0.84; y = −0.0096 x + 3.25). (**D**). Mednorm (B pipeline; *r* = −0.16; the correlation is too low to fit a regression line). ^1^ STAD, ^2^ LUSC, ^3^ LIHC, ^4^ KIRC, ^5^ KIRP, ^6^ BRCA, ^7^ THCA, ^8^ PRAD. The boxes represent the average entropy per cancer type, and the whiskers correspond to their standard deviations.

**Figure 3 biology-13-00482-f003:**
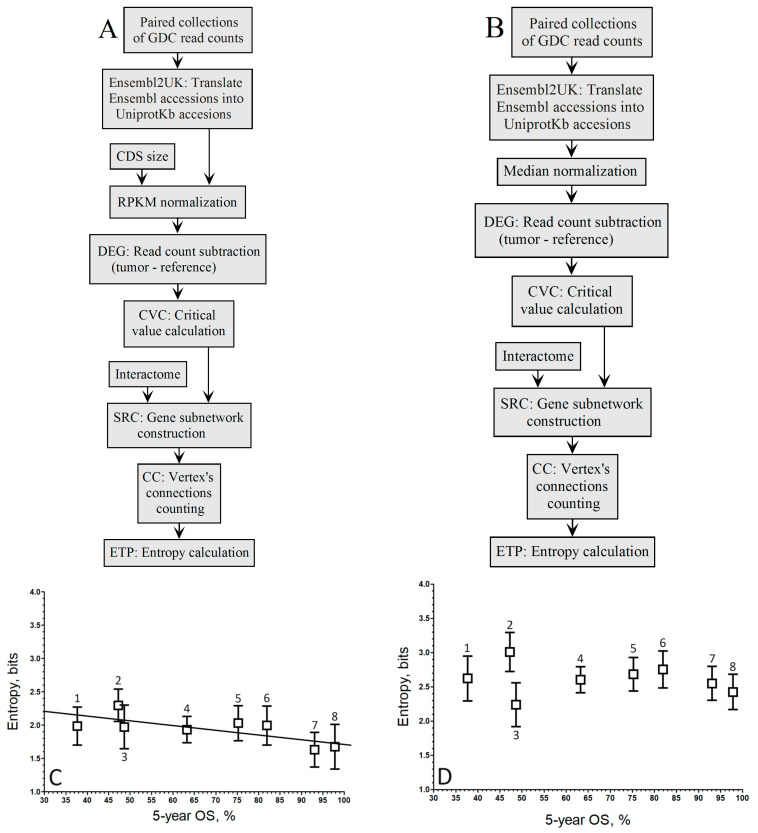
Pipelines to compute the relationship between the entropy of up-regulated gene networks in GDC-paired samples from the RNA-seq of 8 cancer types and their 5-year OS by the *population-wide* approach. (**A**). Pipeline of RPKM normalization without the xlogx step. (**B**). Pipeline of median normalization without the xlogx step. (**C**). RPKM (A pipeline; *r* = −0.72; y = −0.0067 *x* + 2.40). (**D**). Mednorm (B pipeline; *r* = −0.17; the correlation is too low to fit a regression line). ^1^ STAD, ^2^ LUSC, ^3^ LIHC, ^4^ KIRC, ^5^ KIRP, ^6^ BRCA, ^7^ THCA, ^8^ PRAD. The boxes represent the average entropy per cancer type, and the whiskers correspond to their standard deviations.

**Figure 4 biology-13-00482-f004:**
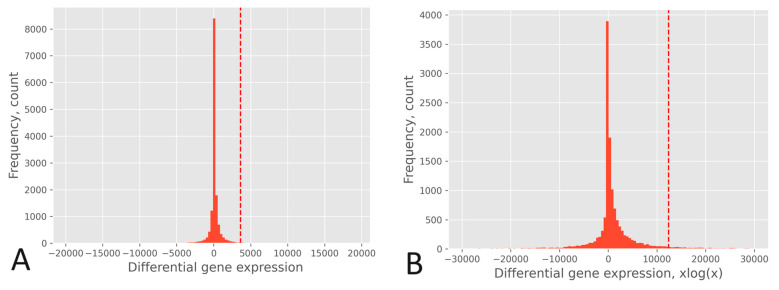
Histogram of the DEGs of the TCGA-A7-A0D9 sample using the RPKM pipeline. (**A**): Without xlogx transformation (pipeline of Figure 3A). (**B**): With xlogx transformation (pipeline of Figure 2A). The dot line stands for the critical value corresponding to the *p*-value 0.025 above which genes are considered up-regulated in tumors relative to their paired control on a *population-wide* basis.

**Figure 5 biology-13-00482-f005:**
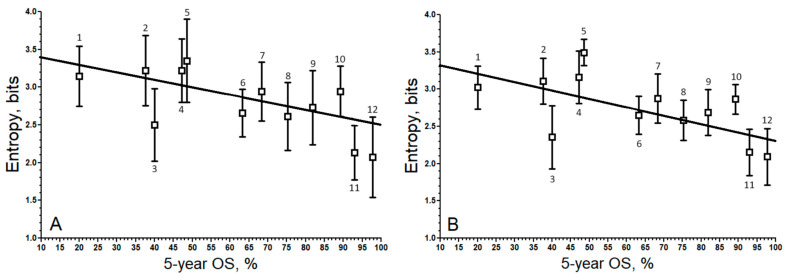
Relationship of degree entropy of the sub-network formed by the up-regulated malignant genes vs. their corresponding patients’ 5 year OS. (**A**). TPM (*r* = −0.67; *y* = −0.0114 *x* + 3.518). (**B**). RLE (*r* = −0.60; *y* = −0.0101 *x* + 3.401). ^1^ BLCA, ^2^ STAD, ^3^ LUAD, ^4^ LUSC, ^5^ LIHC, ^6^ KIRC, ^7^ COAD, ^8^ KIRP, ^9^ BRCA, ^10^ UCS, ^11^ THCA, ^12^ PRAD. The boxes represent the average entropy per cancer type, and the whiskers correspond to their standard deviations.

**Table 1 biology-13-00482-t001:** Raw counts from RNA-seq of paired samples from GDC.

Cancer Type	Abbreviation	OS ^1^	GDC, n ^2^
Bladder carcinoma ^3^	BLCA	20	17
Stomach adenocarcinoma	STAD	38	27
Lung adenocarcinoma	LUAD	40	57
Lung squamous cell carcinoma	LUSC	47	48
Liver hepatocellular carcinoma	LIHC	49	50
Kidney renal clear cell carcinoma	KIRC	63	71
Colon adenocarcinoma ^3^	COAD	68	40
Kidney renal papillary cell carcinoma	KIRP	75	31
Breast cancer	BRCA	82	46
Uterine carcinoma ^3^	UCS	89	22
Thyroid cancer	THCA	93	56
Prostate cancer	PRAD	98	50

^1^ OS: 5-year overall survival taken from Liu et al. [43] according to Conforte et al. [10], %. ^2^ n: sample size, number. ^3^ Accessed on 15 June 2024, see Section 2.2 and Appendix A.

**Table 2 biology-13-00482-t002:** Comparison of RPKM to seven normalization methods and two methods of differentially expressed gene (DEG) identification.

						Normalization Methods					
		DEG Method		NN ^1^		RPKM		TPM		UQ	
Cancer	5-y. OS	DESeq2	edgeR	Av.	St. Dev.	Av.	St. Dev.	Av.	St. Dev.	Av.	St. Dev.
STAD	37.67	2.277	1.452	3.217	0.751	3.114	0.432	3.222	0.468	3.065	0.733
LUSC	47.25	3.028	1.399	3.158	0.451	3.168	0.405	3.221	0.420	2.472	0.619
LIHC	48.63	2.470	1.370	3.452	0.719	3.460	0.551	3.351	0.551	3.178	0.862
KIRC	63.24	2.482	1.224	2.666	0.545	2.671	0.293	2.659	0.318	2.083	0.624
KIRP	75.28	2.406	1.449	2.654	0.612	2.594	0.408	2.613	0.452	2.083	0.611
BRCA	81.90	2.362	1.374	2.812	0.600	2.720	0.468	2.731	0.493	2.377	0.655
THCA	93.02	1.905	1.084	2.354	0.577	2.166	0.340	2.133	0.360	2.002	0.711
PRAD	97.83	1.445	0.738	2.172	0.764	2.097	0.526	2.073	0.534	1.945	0.720
Correl.		−0.722	−0.720	−0.910		−0.914		−0.942		−0.814	
Av.					0.627		0.428		0.449		0.692
St. Dev.					0.109		0.087		0.081		0.084
		**Normalization methods**									
		**Med**		**CPM**		**RLE**		**QN**		**TMM**	
**Cancer**	**5-y. OS**	**Av.**	**St. Dev.**	**Av.**	**St. Dev.**	**Av.**	**St. Dev.**	**Av.**	**St. Dev.**	**Av.**	**St. Dev.**
STAD	37.67	3.100	0.764	3.105	0.433	3.110	0.307	3.217	0.257	3.108	0.319
LUSC	47.25	3.181	0.449	2.487	0.398	3.161	0.354	3.100	0.351	3.162	0.361
LIHC	48.63	3.399	0.733	3.439	0.558	3.495	0.175	3.560	0.208	3.497	0.202
KIRC	63.24	2.690	0.547	2.663	0.291	2.652	0.255	2.641	0.203	2.654	0.244
KIRP	75.28	2.654	0.612	2.588	0.408	2.587	0.270	2.633	0.259	2.588	0.408
BRCA	81.90	2.682	0.599	2.693	0.472	2.690	0.307	2.757	0.307	2.686	0.309
THCA	93.02	2.214	0.561	2.152	0.338	2.154	0.311	2.285	0.259	2.157	0.325
PRAD	97.83	2.046	0.770	2.070	0.528	2.095	0.380	2.150	0.406	2.098	0.379
Correl.		−0.926		−0.777		−0.911		−0.898		−0.911	
Av.			0.629		0.428		0.295		0.281		0.318
St. Dev.			0.116		0.090		0.063		0.070		0.068

^1^ NN stands for “no normalization”.

**Table 3 biology-13-00482-t003:** Comparison of the best performing methods for RNA-seq of 12 cancer types.

		Normalization Methods									
		RPKM		TPM		Med		RLE		TMM	
Cancer	5-y. OS	Av.	St. Dev.	Av.	St. Dev.	Av.	St. Dev.	Av.	St. Dev.	Av.	St. Dev.
BLCA	20.00	3.103	0.368	3.146	0.398	2.847	0.641	3.023	0.288	3.018	0.286
STAD	37.67	3.114	0.432	3.222	0.468	3.100	0.764	3.110	0.307	3.108	0.319
LUAD	40.00	2.500	0.452	2.499	0.479	2.297	0.785	2.355	0.427	2.355	0.429
LUSC	47.25	3.168	0.405	3.221	0.420	3.181	0.449	3.161	0.354	3.162	0.361
LIHC	48.63	3.460	0.551	3.351	0.551	3.399	0.733	3.495	0.175	3.497	0.202
KIRC	63.24	2.671	0.293	2.659	0.318	2.690	0.547	2.652	0.255	2.654	0.244
COAD	68.45	2.943	0.360	2.943	0.390	2.650	1.191	2.878	0.329	2.887	0.308
KIRP	75.28	2.594	0.408	2.613	0.452	2.654	0.612	2.587	0.270	2.588	0.408
BRCA	81.90	2.720	0.468	2.731	0.493	2.682	0.599	2.690	0.307	2.686	0.309
UCS	89.27	2.948	0.285	2.946	0.332	3.323	0.471	2.866	0.197	2.868	0.192
THCA	93.02	2.166	0.340	2.133	0.360	2.214	0.561	2.154	0.311	2.157	0.325
PRAD	97.83	2.097	0.526	2.073	0.534	2.046	0.770	2.095	0.380	2.098	0.379
Correl.		−0.643		−0.674		−0.397		−0.602		−0.598	
Av.			0.407		0.433		0.677		0.300		0.313
St. Dev.			0.084		0.076		0.198		0.071		0.075

## Data Availability

Data are contained within the article or Appendix A.

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
