# Peer review of "Assessing RNA-Seq Workflow Methodologies Using Shannon Entropy"

_biology, 2024, doi:10.3390/biology13070482_

Round 1
Reviewer 1 Report
Comments and Suggestions for Authors
The study focuses on assessing RNA-seq workflow methodologies using Shannon entropy. By analyzing RNA seq datasets from TCGA , the author claimed that RPKM normalization coupled with log2fc yielded the best correlation coefficient of 0.91 between cancer aggressiveness and tumor entropy. Despite the research highlighted the importance of Shannon entropy in evaluating RNA-seq workflows, there are some points remained to be addressed to convince audience for the validity of the findings.
1. Through Shannon entropy was used throughout the manuscript, the validity of the approach in assessing the effectiveness of RNA-seq workflows requires further justification. It would be important for the author to show why it outperforms other metrics in benchmarking all workflows. For examples, the authors should illustrate more on the reason why the entropy serves as the standards for evaluation, which would provide additional insights or validation of the results.
2. The author selected datasets from 475 TCGA RNA-seq datasets across 8 cancer types. The representativeness of these samples and potential biases in the selection process could impact the generalizability of the findings. Analyses of additional datasets with or without batch effects are required to support the findings.
3. The author discussed four types of workflows with difference residing in the normalization. However, these comparisons could be incomprehensive due to that RPKM/FPKM could be probably biased to gene length or high variance between datasets. The author should include other normalization methods including TPM or distribution-driven approaches to support the results. Otherwise, the RPKM with log2fc might not be the best strategies with minimal biases as claimed.
Author Response
- Through Shannon entropy was used throughout the manuscript, the validity of the approach in assessing the effectiveness of RNA-seq workflows requires further justification. It would be important for the author to show why it outperforms other metrics in benchmarking all workflows. For examples, the authors should illustrate more on the reason why the entropy serves as the standards for evaluation, which would provide additional insights or validation of the results.
Answer:
Yes, we agree! We also felt the necessity to justify the entropy as a measure for RNA-seq workflows benchmarking and we took special care to review the literature on this purpose. However, we are not in condition to assess the other metrics in benchmarking. The idea of using entropy for benchmarking proved to be useful and this seemed to us sufficient to be published. To take the remark of referee 1 into account, we propose to add the following section to the beginning of the discussion:
“As noted by Abrams et al. (2019), the primary objective of RNA-seq processing should be the preservation of biological signals. However, given the complexity of biological systems, one must consider how to minimize processing biases while accurately capturing these signals. Although RNA-seq provides information on gene expression, gene expression alone represents only a fraction of biological characteristics and is insufficient to comprehensively account for biological signals. We contend that a system approach to gene expression is necessary to achieve this goal.
Linking RNA-seq data to the interactome provides a system-level dimension to RNA-seq by integrating gene expression data with the topological structure of the biological system under study. The interactome enables for a synthetic conversion of gene expression data into a representation of a biological system, which can serve as a benchmark. However, what are common features of biological systems, and what measures could be used to mathematically represent and assess RNA-seq processing performances?
Among the hallmarks of biological systems, adaptability stands out (Yewdall et al., 2018). There is a tradeoff between adaptability and the robustness of biological networks (Chen and Li, 2010). Robustness indicates a system’s stability, while resilience refers to its ability to return to a stable state after disturbance. Both concepts are fundamental for the survival of biological entities (Zitnik et al., 2019). Numerous studies demonstrated that resilience at lower organization levels contribute to the robustness of entire systems (Crespi et al., 2022). Key features of resilience at the network level include modularity, redundancy, and diversity (Kharrazi et al., 2020), where (i) redundancy in pathways or isoforms enhances the ability to adapt changing environmental conditions, (ii) modularity represents dense clusters of connections between vertices, and (iii) diversity refers to the variety of elements within the system. These features can be depicted by network topology, specifically the distribution of edges per vertex.
Hubs have been linked to protein essentiality based on their positional context in the network (Manke et al., 2006; Biggs et al. 2020). Hubs can also function as bottlenecks, connecting numerous inputs to few outputs. As key connector, these bottleneck proteins are more likely to be essential proteins (Yu et al., 2007). The essentiality of central (modules) hubs is higher than that of local (peripherals) ones (Manke et al., 2006; Biggs et al. 2020), and their inhibitions has a greater disarticulating effect on network compared to lowly connected vertices (Albert et al., 2000; Jeong et al., 2001). Cancer-related hubs may rewire pathways through negative and positive feedback loops (Kitano, 2004; O'Reilly et al., 2006; Chen and Li, 2010; Zitnik et al., 2019; Kennedy et al., 2020; Bergholz and Zhao, 2021; Burkhardt et al., 2022; Crespi et al., 2022) as well as through their ability to connect to partners from different pathways due to their multiple interfaces (Kar et al., 2009). Stochastic changes in the transcriptional programs of tumor cells increase the chances of their survival under any conditions. Gurova (2022) hypothesized that unstable chromatin facilitates stochastic transitions between transcriptional programs in aggressive cancers. These processes also allow the co-option of existing signal transduction pathways for new functions (Kitano, 2004; McLennan, 2008). Many cancer-related hubs are chaperones; they correct misfolded proteins resulting from mutations, thereby decoupling genetic variations (mutations) from phenotypic expression and, thus, isolating low-level variation from high-level functionalities (Zitnik et al., 2019). Such processes of resilience are needed to warrant tumor maintenance or even progression (aggressiveness) in their environment (Kitano, 2004; Cremers et al., 2019).
These considerations highlight the importance of assessing a biological system through networks, particularly PPI networks in the context of cancer. Entropy provides a measure of the structural and dynamic properties of networks, which is biologically significant because the macroscopic resilience of a steady state is correlated with the uncertainty in the underlying microscopic processes, a property measurable by entropy (Manke et al., 2006). Given the cancer-related hubs exhibit topological properties linked to the key biological functions discussed above, their assessment within sub-network of malignant up-regulated genes is essential for uncovering tumor characteristics. The over-expression of hubs significantly contributes to an increase in the overall entropy rate (Teschendorff et al., 2015). However, entropy itself is not very meaningful unless placed in context. Since, patient 5-year overall survival (OS) is a measure of cancer aggressiveness, accurately evaluating its relationship with tumor entropy is of key importance (Conforte et al., 2019).
The biological systems represented by tumors from eight cancer types, encompassing a range of aggressiveness from 30% to 98%, exhibit significant complexity and variance. Sources of variance in RNA-seq include (i) the tissue of sample origin, (ii) sequencing technology, (iii) sequencing equipment, (iv) sequencing professionals, (v) sequencing coverage, (vi) read size, and (vii) differential expression profiles. These factors all affect the workflow performance, raising the question of whether (i) the workflow can achieve similar success rates across different samples with such varying complexity or (ii) it performs better with specific sample types. Ideally, a workflow that can provide suitable outputs over a broad range of complexity is desired. Maximizing the correlation coefficient between entropy and aggressiveness across this range of biological complexity is equivalent to identifying the combination of technologies that performs best across the variance spectrum.
It is well recognized that multiple genomic clonal populations within a neoplasm arise from divergent evolution of the originating cell progeny, increasing tumor heterogeneity over time (El-Deiry et al., 2017; La Rosa et al., 2019; Janiszewska, 2020; Martínez-Gregorio et al., 2021; Choi et al., 2023; García-Montaño et al., 2023). Significant temporal changes in transcriptome profiling and chromatin accessibility have been observed as a result of the emergence of distinct cell populations (Brady et al., 2021).
Since tumor aggressiveness is recognized to be positively correlated to complexity (Mullins et al., 2012; Ciriello et al., 2013; Kwon et al., 2019; Kalasekar et al., 2021), entropy can be concluded to be a suitable measure for benchmarking RNA-seq workflows. Here, we consider tumor heterogeneity and complexity as a whole, given that we only addressed bulk RNA-seq. In bulk RNA-seq, up-regulated genes represent the average expression levels conserved across the different cell lineages within the tumor sample being sequenced (Wang et al., 2019). If this sample is representative of the entire tumor, one may hypothesize that up-regulated genes identified in bulk RNA-seq represent the primary determinants of cancer; those that render the tumor compatible with its environment. Expanding on this, the quantitative concept of aggressiveness can be defined as a secondary determinant of biological systems, stemming from their ability to effectively exploit an ecological niche (Yoder, 1980). In the context of cancer, this niche exists between the tumor and its host tissue. In contrast to genes whose over-expression is consistently significant throughout the tumor, those that are selectively over-expressed in specific cell lineages without a discernible impact on the overall tumor level could be regarded as secondary determinants.
As observed by Baltazar et al. (2019), “mean entropies represent the average contribution from individual hubs”. Therefore, if aggressive tumors exhibit greater complexity and heterogeneity on a molecular level, and if networks accurately represent tumor biology, entropy becomes a suitable measure of network topology (i.e., complexity) and aggressiveness, as demonstrated by Conforte et al. (2019). Moreover, hubs should be recognized as key components driving network entropy (Albert et al., 2000). Hence, we advocate for the relevance of utilizing the relationship between tumoral entropy and 5-year OS as a benchmark for the evaluating RNA-seq processing methodologies. At least, we recognized the necessity of identifying the best workflow for diagnosing the hubs that would be most suitable for theranostic purposes.”
Reference
Abrams, Z.B.; Johnson, T.S.; Huang, K.; Payne, P.R.O.; Coombes, K. A protocol to evaluate RNA sequencing normalization methods. BMC Bioinformatics 2019, 20, 679. doi: 10.1186/s12859-019-3247-x.
Albert, R.; Jeong, H.; Barabási, A.L. Error and attack tolerance of complex networks. Nature 2000, 406, 378-382. doi: 10.1038/35019019.
Baltazar, C.A.; Guinle, M.I.B.; Caron, C.J.; Amaro, Jr.E.; Machado, B.S. Connective core structures in cognitive networks: The role of hubs. Entropy 2019, 21, 961. doi: 10.3390/e21100961.
Bergholz, J.S.; Zhao, J.J. How compensatory mechanisms and adaptive rewiring have shaped our understanding of therapeutic resistance in cancer. Cancer Res. 2021, 81, 6074-6077. doi: 10.1158/0008-5472.CAN-21-3605.
Biggs, C.R.; Yeager, L.A.; Bolser, D.G.; Bonsell, C.; Dichiera, A.M.; Hou, Z.; Keyser, S.R.; Khursigara, A.J.; Lu, K.; Muth, A.F.; et al. Does functional redundancy affect ecological stability and resilience? A review and meta-analysis. Ecosphere 2020, 11, e03184. 10.1002/ecs2.3184.
Brady, N.J.; Bagadion, A.M.; Singh, R.; Conteduca, V.; Van Emmenis, L.; Arceci, E.; Pakula, H.; Carelli, R.; Khani, F.; Bakht, M.; et al. Temporal evolution of cellular heterogeneity during the progression to advanced AR-negative prostate cancer. Nat. Commun. 2021, 12, 3372. doi: 10.1038/s41467-021-23780-y
Burkhardt, D.B.; San Juan, B.P.; Lock, J.G.; Krishnaswamy, S.; Chaffer, C.L. Mapping phenotypic plasticity upon the cancer cell state landscape using manifold learning. Cancer Discov. 2022, 12, 1847-1859. doi: 10.1158/2159-8290.CD-21-0282.
Chen, B.-S.; Li, C.-W. On the interplay between entropy and robustness of gene regulatory networks. Entropy 2010, 12, 1071-1101. doi: 10.3390/e12051071.
Choi, J.H.; Lee, B.S.; Jang, J.Y.; Lee, Y.S.; Kim, H.J.; Roh, J.; Shin, Y.S.; Woo, H.G.; Kim, C.H. et al. Single-cell transcriptome profiling of the stepwise progression of head and neck cancer. Nat. Commun. 2023, 14, 1055. doi: 10.1038/s41467-023-36691-x
Ciriello, G.; Miller, M.L.; Aksoy, B.A.; Senbabaoglu, Y.; Schultz, N.; Sander, C. Emerging landscape of oncogenic signatures across human cancers. Nat. Genet. 2013; 45, 1127-1133. doi:10.1038/ng.2762.
Conforte, A.J.; Tuszynski, J.A.; da Silva, F.A.B.; Carels, N. Signaling complexity measured by Shannon entropy and its application in personalized medicine. Front. Genet. 2019, 10, 930. doi: 10.3389/fgene.2019.00930.
Cremers, C.G.; Nguyen, L.K. Network rewiring, adaptive resistance and combating strategies in breast cancer. Cancer Drug. Resist. 2019, 19, 1106-1126. doi: 10.20517/cdr.2019.60.
Crespi, E.; Burnap, R.; Chen, J.; Das, M.; Gassman, N.; Rosa, E.; Simmons, R.; Wada, H.; Wang, Z.Q.; Xiao, J.; et al. Resolving the rules of robustness and resilience in biology across scales. Integr. Comp. Biol. 2022, 61, 2163-2179. doi: 10.1093/icb/icab183.
El-Deiry, W.S.; Taylor, B.; Neal, J.W. Tumor evolution, heterogeneity, and therapy for our patients with advanced cancer: How far have we come? Am. Soc. Clin. Oncol. Educ. Book. 2017, 37, e8-e15. doi: 10.1200/EDBK_175524.
García-Montaño, L.A.; Licón-Muñoz, Y.; Martinez, F.J.; Keddari, Y.R.; Ziemke, M.K.; Chohan, M.O.; Piccirillo, S.G.M. Dissecting intra-tumor heterogeneity in the glioblastoma microenvironment using fluorescence-guided multiple sampling. Mol. Cancer Res. 2023, 21, 755-767. doi: 10.1158/1541-7786.MCR-23-0048.
Gurova, K. Can aggressive cancers be identified by the "aggressiveness" of their chromatin? Bioessays 2022, 44, e2100212. doi: 10.1002/bies.202100212.
Hu, G.; Wu, Z.; Uversky, V.N.; Kurgan, L.; Functional analysis of human hub proteins and their interactors involved in the intrinsic disorder-enriched interactions. Int. J. Mol. Sci. 2017, 18, 2761. doi: 10.3390/ijms18122761.
Janiszewska, M. The microcosmos of intratumor heterogeneity: the space-time of cancer evolution. Oncogene 2020, 39, 2031–2039. doi: 10.1038/s41388-019-1127-5.
Jeong, H.; Mason, S.P.; Barabási, A.L.; Oltvai, Z.N. Lethality and centrality in protein networks. Nature 2001, 411, 41-42. Doi: 10.1038/35075138.
Kalasekar, S.M.; VanSant-Webb, C.H.; Evason, K.J. Intratumor heterogeneity in hepatocellular carcinoma: challenges and opportunities. Cancers 2021, 13, 5524. doi: 10.3390/cancers13215524.
Kar, G.; Gursoy, A.; Keskin, O. Human cancer protein-protein interaction network: A structural perspective. PLoS Comput. Biol. 2009, 5, e1000601. doi: 10.1371/journal.pcbi.1000601.
Kennedy, S.A.; Jarboui, M.A.; Srihari, S.; Raso, C.; Bryan, K.; Dernayka, L.; Charitou, T.; Bernal-Llinares, M.; Herrera-Montavez, C.; Krstic, A.; et al. Extensive rewiring of the EGFR network in colorectal cancer cells expressing transforming levels of KRASG13D. Nat. Commun. 2020, 11, 499. doi: 10.1038/s41467-019-14224-9.
Kharrazi, A.; Yu, Y.; Jacob, A.; Vora, N.; Fath, B.D. Redundancy, diversity, and modularity in network resilience: Applications for international trade and implications for public policy. Curr. Res. Environ. Sustain. 2020, 2, 100006. doi: 10.1016/j.crsust.2020.06.001.
Kitano, H. Biological robustness. Nat. Rev. Genet. 2004, 5, 826-837. doi: 10.1038/nrg147
Kwon, S.M.; Budhu, A.; Woo, H.G.; Chaisaingmongkol, J.; Dang, H.; Forgues, M.; Harris, C.C.; Zhang, G.; Auslander, N.; Ruppin, E.; et al. Functional genomic complexity defines intratumor heterogeneity and tumor aggressiveness in liver cancer. Sci. Rep. 2019, 9, 16930. doi: 10.1038/s41598-019-52578-8.
La Rosa, S.; Rubbia-Brandt, L.; Scoazec, J.-Y.; Weber, A. Editorial: Tumor Heterogeneity. Front. Med. 2019, 6. doi: 10.3389/fmed.2019.00156
Manke, T.; Demetrius, L.; Vingron, M. An entropic characterization of protein interaction networks and cellular robustness. J. R. Soc. Interface 2006, 3, 843-850. doi: 10.1098/rsif.2006.0140.
Martínez-Gregorio, H.; Rojas-Jiménez, E.; Mejía-Gómez, J.C.; Díaz-Velásquez, C.; Quezada-Urban, R.; Vallejo-Lecuona, F.; de la Cruz-Montoya, A.; Porras-Reyes, F.I.; Pérez-Sánchez, V.M.; Maldonado-Martínez, H.A.; et al. The evolution of clinically aggressive triple-negative breast cancer shows a large mutational diversity and early metastasis to lymph nodes. Cancers 2021, 13, 5091. doi: 10.3390/cancers13205091.
McLennan, D.A. The concept of co-option: Why evolution often looks miraculous. Evo. Edu. Outreach. 2008, 1, 247–258. doi: 10.1007/s12052-008-0053-8.
Mullins, J.K.; Kaouk, J.H.; Bhayani, S.; Rogers, C.G.; Stifelman, M.D.; Pierorazio, P.M.; Tanagho, Y.S.; Hillyer, S.P.; Kaczmarek, B.F.; Chiu, Y.; Allaf, M.E. Tumor complexity predicts malignant disease for small renal masses. J. Urol. 2012, 188, 2072-2076. doi: 10.1016/j.juro.2012.08.027.
O'Reilly, K.E., Rojo, F., She, Q.B.; Solit, D.; Mills, G.B.; Smith, D.; Lane, H.; Hofmann, F.; Hicklin, D.J.; Ludwig, D.L.; et al. mTOR inhibition induces upstream receptor tyrosine kinase signaling and activates Akt. Cancer Res. 2006, 66, 1500-1508. doi: 10.1158/0008-5472.CAN-05-2925.
Teschendorff, A.E.; Banerji, C.R.; Severini, S.; Kuehn, R.; Sollich, P. Increased signaling entropy in cancer requires the scale-free property of protein interaction networks. Sci. Rep. 2015, 5, 9646. doi: 10.1038/srep09646.
Wang, X.; Park, J.; Susztak, K.; Zhang, NR.; Li, M. Bulk tissue cell type deconvolution with multi-subject single-cell expression reference. Nat. Commun. 2019, 10, 380. doi: 10.1038/s41467-018-08023-x.
Yewdall, N.A.; Mason, A.F.; van Hest, J.C.M. The hallmarks of living systems: towards creating artificial cells. Interface Focus. 2018, 8, 20180023. doi: 10.1098/rsfs.2018.0023.
Yoder, O.C. Toxins in pathogenesis. Annu. Rev. Phythopathol. 1980, 18, 103–129. doi: 10.1146/annurev.py.18.090180.000535.
Yu, H.; Kim, P.M.; Sprecher, E.; Trifonov, V.; Gerstein, M. The importance of bottlenecks in protein networks: correlation with gene essentiality and expression dynamics. PLoS Comput. Biol. 2007, 3, e59. doi:10.1371/journal.pcbi.0030059.
Zitnik, M.; Sosič, R.; Feldman, M.W.; Leskovec, J. Evolution of resilience in protein interactomes across the tree of life. Proc. Natl. Acad. Sci. USA 2019, 116, 4426-4433. doi: 10.1073/pnas.1818013116.
- The author discussed four types of workflows with difference residing in the normalization. However, these comparisons could be incomprehensive due to that RPKM/FPKM could be probably biased to gene length or high variance between datasets. The author should include other normalization methods including TPM or distribution-driven approaches to support the results. Otherwise, the RPKM with log2fc might not be the best strategies with minimal biases as claimed.
Answer
Yes, we agree! The following observations #2 and #3 are typical of Bayesian learning. To attend these observations, I divided the manuscript in three steps:
Step 1 is populated with the results of the previous version as there was no remark about them.
Step 2 is populated by comparison of RKM with DESeq2, and edger as well as TPM, UQ, Med, CPM, RLE, QN, TMM in combination with log2 fold change.
Step 3, we considered the results of Step 2 as a calibration step or a training step. Based on the best performing methodology identified in Step 2, we tested additional paired samples (BCLA, LUAD, COAD, and UCS) within the correlation between tumoral entropy and 5-year OS. This approach is presented at the end of the introduction with the following paragraph between lines 142 and 168 in the introduction:
“We validated this process in three steps: (i)Accordingly, across 8 types of cancers (475 patients) spanning 5-year OS rates from 30 % to 98%, We evaluated the performance of RPKM and Median normalization on a gene-by-gene basis or and by referencing the population of DEGs across 8 types of cancers (475 patients) spanning 5-year OS rates from 30 % to 98% according to previous studies (Conforte et al., 2019; Pires et al., 2021). (ii) We compared RPKM to seven read count normalizations, i.e., Transcript Per Million (TPM, Li and Dewey, 2011), Counts Per Million (CPM, Dillies et al., 2013), Median (Med, Dillies et al., 2013), Upper Quantile (UQ, Bullard et al. 2010), Relative log expression (RLE, Love et al., 2014), Quantile normalization (QN, Bolstad et al., 2003), and Trimmed mean of M-values (TMM, Robinson and Oshlack, 2010) as well as to two cross-sample distribution based methods, i.e., DESeq2 (Love et al. 2014) and edgeR (Robinson et al. 2010). (iii) Based on the best performing methodologies identified in these comparison, we tested paired samples from additional cancer types from TCGA to validate the approach, including bladder carcinoma (BLCA), lung adenocarcinoma (LUAD), colon adenocarcinoma (COAD), and uterine carcinoma (UCS).
According to this Bayesian learning process, we found the following: (i) We used tThe coefficient of correlation between average entropy per cancer type and aggressiveness (5-year OS) ais a suitable metric for the comparative performance of biological information extraction of sub-networks of up-regulated malignant genes. In our hands, (ii) TMM, QN, and RLE, a group of methods that determine a scaling factor for variation stabilization, produced a correlation coefficient similar to TPM, but with a standard deviation approximately 25% lower. The straightforward approach of combining a the RPKM normalization method (even without a scaling factor for variation stabilization) with log2 fold change yielded better average correlation coefficients than probabilistic methods such as DESeq2 and edgeR for determining network entropy. associated to the log2 fold change was the process that gave the most consistent results in terms of entropy.(iii) The correlation coefficient decreased from r ≈ 0.9 to r ≈ 0.6 when the number of cancer types increased from 8 to 12.”
Conforte AJ, Tuszynski JA, da Silva FAB, Carels N (2019) Signaling complexity measured by Shannon entropy and its application in personalized medicine. Front Genet 10, 930.
Pires JG, da Silva GF, Weyssow T, Conforte AJ, Pagnoncelli D, da Silva FAB, Carels N (2021) Galaxy and MEAN Stack to create a user-friendly workflow for the rational optimization of cancer chemotherapy. Front Genet 12, 624259.
Li, B.; Dewey, C.N. RSEM: accurate transcript quantification from RNA-Seq data with or without a reference genome. BMC Bioinformatics 2011, 12, 323. doi: 10.1186/1471-2105-12-323.
Dillies, M.-A.; Rau, A.; Aubert J.; Hennequet-Antier, C.; Jeanmougin, M.; Servant, N.; Keime, C.; Marot G.; Castel D.; Estelle, J.; et al. A comprehensive evaluation of normalization methods for Illumina high-throughput RNA sequencing data analysis. Briefings Bioinf. 2013, 14, 671–683. doi: 10.1093/bib/bbs046.
Bullard, J.H.; Purdom, E.; Hansen, K.D.; Dudoit, S. Evaluation of statistical methods for normalization and differential expression in mRNA-Seq experiments. BMC Bioinf. 2010, 11, 94. doi: 10.1186/1471-2105-11-94.
Bolstad, B.M.; Irizarry, R.A.; Åstrand, M.; Speed, T.P. A comparison of normalization methods for high density oligonucleotide array data based on variance and bias. Bioinformatics 2003, 19, 185–193. doi: 10.1093/bioinformatics/19.2.185.
Robinson, M.D.; Oshlack, A. A scaling normalization method for differential expression analysis of RNA-seq data. Genome Biol. 2010, 11, R25. doi: 10.1186/gb-2010-11-3-r25.
Robinson, M.D.; McCarthy, D.J.; & Smyth, G.K. edgeR: a Bioconductor package for differential expression analysis of digital gene expression data. Bioinformatics 2010, 26, 139-140. doi: 10.1093/bioinformatics/btp616.
In the Materials and Method section, we added:
“TPM: The gene expression in transcripts per million (TPM) of a gene is defined as the ratio of it RPK over the sum of all RPKs (per million) [38]. We computed raw counts in accordance with formula 2 using a custom Perl script derived from the one used for RPKM calculation.
2.4. Extended normalization methods
In this section, we aim to compare the normalization methods provided by the NormSeq server (https://arn.ugr.es/normSeq) to RPKM and TPM. Briefly and citing Scheepbouwer et al. (2023), the purpose of these methods can be summarized as follow:
Counts Per Million (CPM): CPM normalization corrects for library size without considering transcript length (Dillies et al., 2013).
Median (Med): Median normalization adjusts the data of each individual sample by adding a constant value to achieve the same median value across all samples (Dillies et al., 2013).
Upper Quantile (UQ): All genes with a read count of 0 are removed, followed by a division of the remaining gene counts by the upper quartile (Bullard et al. 2010).
Relative log expression (RLE): For each gene, the RLE scaling factor is computed as the median of the ratio of the read counts by taking the geometric mean across all samples (Love et al., 2014).
Quantile normalization (QN): Quantile normalization applies a mathematical transformation to the rank statistics across samples (Bolstad et al., 2003).
Trimmed mean of M-values (TMM): The TMM method estimates scale factors for comparing libraries on a relative scale (Robinson and Oshlack, 2010).
2.5. Differential expression method
Here we considered DESeq2 (Love et al. 2014) and edgeR (Robinson et al. 2010) as reference software for benchmarking the capacity of entropy to report on the extraction of biological information given the complexity of sub-networks associated with malignant up-regulated genes. Both packages are cross-sample distribution-based methods that estimate the dispersion parameter for each gene, reflecting the variability of read counts according to the negative binomial distribution. These software apply the Benjamini-Hochberg procedure to control the false discovery rate (FDR), helping manage the multiple testing problem inherent in RNA-seq data analysis. However, they differ in their normalization methods: DESeq2 is based on the median of ratios to normalize read counts (MRN) and EdgeR is based on trimmed mean of M-values (TMM). DESeq2 was run from the iDEP server (http://bioinformatics.sdstate.edu/idep/) (Ge et al., 2018) and edgeR from NormSeq (https://arn.ugr.es/normSeq).
”
In the Results section, we added the following data:
“3.2. Step 2: Comparison of normalization methods and differential expression determination processes
The results in Table 2 were produced by replacing RPKM by TPM in the pipeline of Figure 1A and replacing the normalization method step in Figure 1B with UQ, Med, CPM, RLE, QN, or TMM (Table S3). Table 2 shows that the correlation coefficient between entropy and 5-year OS for the sub-networks of malignant up-regulated genes of eight cancer types was lower for DESeq2 and edgeR (r = 0.72) compared to the correlation obtained with log2 fold change ≥ 1 filter applied to raw count normalized with any methods (even with no normalization). RLE and PCA plots are given in Figure S2 for DESeq2.
Table 2. Comparison of RPKM to seven normalization methods and two differentially expressed methods (DEG).
|
|
|
|
|
|
|
Normalization methods |
||||||
|
|
|
DEG method |
NN1 |
|
RPKM |
|
TPM |
|
UQ |
|
||
|
Cancer |
5-y. OS |
DESeq2 |
edgeR |
Av. |
St. Dev. |
Av. |
St. Dev. |
Av. |
St. Dev. |
Av. |
St. Dev. |
|
|
STAD |
37.67 |
2.277 |
1.452 |
3.217 |
0.751 |
3.114 |
0.432 |
3.222 |
0.468 |
3.065 |
0.733 |
|
|
LUSC |
47.25 |
3.028 |
1.399 |
3.158 |
0.451 |
3.168 |
0.405 |
3.221 |
0.420 |
2.472 |
0.619 |
|
|
LIHC |
48.63 |
2.470 |
1.370 |
3.452 |
0.719 |
3.460 |
0.551 |
3.351 |
0.551 |
3.178 |
0.862 |
|
|
KIRC |
63.24 |
2.482 |
1.224 |
2.666 |
0.545 |
2.671 |
0.293 |
2.659 |
0.318 |
2.083 |
0.624 |
|
|
KIRP |
75.28 |
2.406 |
1.449 |
2.654 |
0.612 |
2.594 |
0.408 |
2.613 |
0.452 |
2.083 |
0.611 |
|
|
BRCA |
81.90 |
2.362 |
1.374 |
2.812 |
0.600 |
2.720 |
0.468 |
2.731 |
0.493 |
2.377 |
0.655 |
|
|
THCA |
93.02 |
1.905 |
1.084 |
2.354 |
0.577 |
2.166 |
0.340 |
2.133 |
0.360 |
2.002 |
0.711 |
|
|
PRAD |
97.83 |
1.445 |
0.738 |
2.172 |
0.764 |
2.097 |
0.526 |
2.073 |
0.534 |
1.945 |
0.720 |
|
|
|
|
|
|
|
|
|
|
|
|
|
|
|
|
Correl. |
|
-0.722 |
-0.720 |
-0.910 |
|
-0.914 |
|
-0.942 |
|
-0.814 |
|
|
|
Av. |
|
|
|
|
0.627 |
|
0.428 |
|
0.449 |
|
0.692 |
|
|
St. Dev. |
|
|
|
|
0.109 |
|
0.087 |
|
0.081 |
|
0.084 |
|
1NN stands for “no normalization”.
Table 2 (continued). Comparison of RPKM to seven normalization methods and two DEG methods.
|
|
|
Normalization methods |
|||||||||
|
|
|
Med |
|
CPM |
|
RLE |
|
QN |
|
TMM |
|
|
Cancer |
5-y. OS |
Av. |
St. Dev. |
Av. |
St. Dev. |
Av. |
St. Dev. |
Av. |
St. Dev. |
Av. |
St. Dev. |
|
STAD |
37.67 |
3.100 |
0.764 |
3.105 |
0.433 |
3.110 |
0.307 |
3.217 |
0.257 |
3.108 |
0.319 |
|
LUSC |
47.25 |
3.181 |
0.449 |
2.487 |
0.398 |
3.161 |
0.354 |
3.100 |
0.351 |
3.162 |
0.361 |
|
LIHC |
48.63 |
3.399 |
0.733 |
3.439 |
0.558 |
3.495 |
0.175 |
3.560 |
0.208 |
3.497 |
0.202 |
|
KIRC |
63.24 |
2.690 |
0.547 |
2.663 |
0.291 |
2.652 |
0.255 |
2.641 |
0.203 |
2.654 |
0.244 |
|
KIRP |
75.28 |
2.654 |
0.612 |
2.588 |
0.408 |
2.587 |
0.270 |
2.633 |
0.259 |
2.588 |
0.408 |
|
BRCA |
81.90 |
2.682 |
0.599 |
2.693 |
0.472 |
2.690 |
0.307 |
2.757 |
0.307 |
2.686 |
0.309 |
|
THCA |
93.02 |
2.214 |
0.561 |
2.152 |
0.338 |
2.154 |
0.311 |
2.285 |
0.259 |
2.157 |
0.325 |
|
PRAD |
97.83 |
2.046 |
0.770 |
2.070 |
0.528 |
2.095 |
0.380 |
2.150 |
0.406 |
2.098 |
0.379 |
|
|
|
|
|
|
|
|
|
|
|
|
|
|
Correl. |
|
-0.926 |
|
-0.777 |
|
-0.911 |
|
-0.898 |
|
-0.911 |
|
|
Av. |
|
|
0.629 |
|
0.428 |
|
0.295 |
|
0.281 |
|
0.318 |
|
St. Dev. |
|
|
0.116 |
|
0.090 |
|
0.063 |
|
0.070 |
|
0.068 |
This table also shows that TPM (r = 0.94) ranks highest among methods without variance stabilization by a scaling factor, such as RPKM (r = 0.91), UQ (r = 0.81), Med (r = 93), and CPM (r = 0.77). Some of these methods performed even better than variance-stabilized methods (RLE: r = 0.91, QN: r = 0.90, TMM: r = 0.91), but they exhibited nearly double the rate of average standard deviation (0.4 to 0.6 compared to ~0.3).
The variance stabilization of RLE, QN, and TMM normalizations can be verified from their RLE plots (Gandolfo, 2018) compared to TPM, RPKM, UQ, Med, and CPM (Figures S3). However, variance stabilization does not necessarily improved the PCA classification of control and tumor samples (Figures S4); in some cases PCA classification is effective, and in other, it is not, without any apparent correlation to any specific feature.
Gandolfo, L.C.; Speed, T.P. RLE plots: Visualizing unwanted variation in high dimensional data. PLoS One 2018, 13, e0191629. doi: 10.1371/journal.pone.0191629. “
In the Discuttion section, we added:
“4.3. Comparison of normalization methods and differential expression determination processes
Interestingly we found that DESeq2 and edgeR were not effective in assessing the complexity of biological network across a large cohort encompassing a 5-year OS aggressiveness rate ranging from 20 to 98%. From this finding, one might conclude that methods relying on negative binomial distribution and Benjamini-Hochberg procedure for false discovery rate (FDR) control are not optimal for extracting biological complexity from RNA-seq data. Conversely, normalization methods such as TPM, RLE, and TMM combined with log2 fold change appear suitable for this purpose.”
- The author selected datasets from 475 TCGA RNA-seq datasets across 8 cancer types. The representativeness of these samples and potential biases in the selection process could impact the generalizability of the findings. Analyses of additional datasets with or without batch effects are required to support the findings.
Answer
Yes, we agree! We extend the sample size of this study with four additional cancer types: BLCA, LUAD, COAD, USC.
In the Result section, we added:
“3.3. Step 3: Generalization of the degree-entropy vs. 5-year OS relationship
Since the eight cancer type in Table 2 could be the result of an over-fitting process, we include paired samples of four additional cancer types: BLCA, LUAD, COAD, and UCS. Table 3 shows that it is indeed the case; however, the negative relationship between entropy and 5-year OS is maintained. The best result was obtained with TPM (r = –0.674), while RLE (r = –0.602) and TMMM (r = –0.598) exhibited similar lower correlation coefficients but also lower variance.
Table 3. Comparison of the best performing methods for RNA-seq of 12 cancer types.
|
|
|
Normalization methods |
|
|
|
|
|
|
|||
|
|
|
RPKM |
|
TPM |
|
Med |
|
RLE |
|
TMM |
|
|
Cancer |
5-y. OS |
Av. |
St. Dev. |
Av. |
St. Dev. |
Av. |
St. Dev. |
Av. |
St. Dev. |
Av. |
St. Dev. |
|
BLCA |
20.00 |
3.103 |
0.368 |
3.146 |
0.398 |
2.847 |
0.641 |
3.023 |
0.288 |
3.018 |
0.286 |
|
STAD |
37.67 |
3.114 |
0.432 |
3.222 |
0.468 |
3.100 |
0.764 |
3.110 |
0.307 |
3.108 |
0.319 |
|
LUAD |
40.00 |
2.500 |
0.452 |
2.499 |
0.479 |
2.297 |
0.785 |
2.355 |
0.427 |
2.355 |
0.429 |
|
LUSC |
47.25 |
3.168 |
0.405 |
3.221 |
0.420 |
3.181 |
0.449 |
3.161 |
0.354 |
3.162 |
0.361 |
|
LIHC |
48.63 |
3.460 |
0.551 |
3.351 |
0.551 |
3.399 |
0.733 |
3.495 |
0.175 |
3.497 |
0.202 |
|
KIRC |
63.24 |
2.671 |
0.293 |
2.659 |
0.318 |
2.690 |
0.547 |
2.652 |
0.255 |
2.654 |
0.244 |
|
COAD |
68.45 |
2.943 |
0.360 |
2.943 |
0.390 |
2.650 |
1.191 |
2.878 |
0.329 |
2.887 |
0.308 |
|
KIRP |
75.28 |
2.594 |
0.408 |
2.613 |
0.452 |
2.654 |
0.612 |
2.587 |
0.270 |
2.588 |
0.408 |
|
BRCA |
81.90 |
2.720 |
0.468 |
2.731 |
0.493 |
2.682 |
0.599 |
2.690 |
0.307 |
2.686 |
0.309 |
|
UCS |
89.27 |
2.948 |
0.285 |
2.946 |
0.332 |
3.323 |
0.471 |
2.866 |
0.197 |
2.868 |
0.192 |
|
THCA |
93.02 |
2.166 |
0.340 |
2.133 |
0.360 |
2.214 |
0.561 |
2.154 |
0.311 |
2.157 |
0.325 |
|
PRAD |
97.83 |
2.097 |
0.526 |
2.073 |
0.534 |
2.046 |
0.770 |
2.095 |
0.380 |
2.098 |
0.379 |
|
|
|
|
|
|
|
|
|
|
|
|
|
|
Correl. |
|
-0.643 |
|
-0.674 |
|
-0.397 |
|
-0.602 |
|
-0.598 |
|
|
Av. |
|
|
0.407 |
|
0.433 |
|
0.677 |
|
0.300 |
|
0.313 |
|
St. Dev. |
|
|
0.084 |
|
0.076 |
|
0.198 |
|
0.071 |
|
0.075 |
By plotting the relationship entropy vs 5-year OS for TPM (Figure 5A) and RLE (Figure 5B), one can better visualize the lower variation associated to RLE compared to TPM. Both relationships are very similar. The decrease in the correlation coefficient from TPM to RLE or TMM is due to the decrease in covariance. The covariance for TPM was –6.458 while for RLE, it was –5763. From this, one may conclude that variance stabilization through the application of a scaling factor has a negative effect on the correlation.
Fig. 5. Relationship of degree entropy of the sub-network formed by the up-regulated malignant genes vs their corresponding patients’ 5year OS. A. TPM (r = –0.67; y=–0.0114*x+3.518). B. RLE (r = –0.60; y=–0.0101*x+3.401). 1BLCA, 2STAD, 3LUAD, 4LUSC, 5LIHC, 6KIRC, 7COAD, 8KIRP, 9BRCA, 10UCS, 11THCA, 12PRAD. The boxes represent the average entropy per cancer type and the whiskers correspond to their standard deviations.”
In the Discussion section, we added:
4.4. The relationship of degree-entropy and 5-year OS
It is interesting to note that the negative correlation between degree-entropy and 5-year OS, albeit this correlation diminishes notably when analyzing paired samples of 12 cancer types rather than eight. The underperformance of Med in this context indicates its inadequate adaptation to biological samples characterized by significant topological complexity variation. This underscores the presence of over-fitting in the previous analyses. Nevertheless, the observed reduction in correlation remains intriguing as it suggests that tumors adjusted their gene up-regulation patterns to form more o less complex sub-networks depending on the cancer type and its environment. This consideration holds importance in guiding clinical decisions regarding optimal therapeutic strategies. For instance, therapies targeting hubs may be less effective in tumors exhibiting lower entropy levels.
Among the types of errors that could account for the correlation reduction across the 12 cancer types, it should be noted that the standard deviation associated with the x axis is unknown. Although 5-year OS is a statistical metric intended to ensure robustness of the data on the x axis, uncertainties remain. Another factor to consider is that the efficacy of treatments for specific cancer type can vary independently of their aggressiveness. This variability can influence the 5-year OS, which have generally increased over time but at differing rates depending on the type of cancer. “
Reviewer 2 Report
Comments and Suggestions for Authors
Nicolas Carels found that the correlation between sub-network entropy and 5-year overall survival rates can serve as a benchmark for optimizing RNA-seq workflows. The author identified that the pipeline incorporating RPKM normalization coupled with log2 fold change performs best. The paper can be improved with the following aspects:
-
In the methods section, it is still unclear how entropy is calculated using a PPI network and how PPI data is obtained from up-regulated genes. Did the author use custom scripts or Perl packages? I suggest the author clarify this.
-
Are figures 1CD, 2CD, and 3CD boxplots? I suggest the author add figure legends explaining what the box and whisker represent in the plots.
-
DESeq2 and edgeR are the most widely used DEGs pipelines in this area; the author should benchmark these two tools.
-
The author mentioned that cancer-enriched PPI networks are more redundant in aggressive cancers. Therefore, the entropy-survival rates correlation benchmark methods can only be used when analyzing other cancer-related RNA-seq data, not all RNA-seq analyses. The author should clarify this in the paper.
Author Response
Referee 2:
Comments and Suggestions for Authors:
- In the methods section, it is still unclear how entropy is calculated using a PPI network and how PPI data is obtained from up-regulated genes. Did the author use custom scripts or Perl packages? I suggest the author clarify this.
Answer:
Yes, we agree! All the scripts of this report are custom and are give in the GitHub link at section 2.5. We now better detailed the process of sub-network establishment, edge counting, and entropy calculation as follow:
“The edges between sub-network vertices were established by reference to the interactions described in the IntAct interactome (Orchard et al., 2014). Then, the sub-network vertices were listed and counted. Finally, the events of k edges were computed from the minimum value of one edge between two vertices to the event corresponding to the vertex with the maximal edges n with its neighbors for the sub-network under consideration. When k did not match any vertex in the network, its corresponding entropy was not computed because it would result in an undefined value (log2(0) is undefined). All the Perl scripts involved in this process were custom and described in Pires et al. [11]). The script for entropy calculation produced results identical to those obtained using the Entropy function of R (https://www.rdocumentation.org/packages/DescTools/versions/0.99.54/topics/Entropy).”
Orchard, S.; Ammari, M.; Aranda, B.; Breuza, L.; Briganti, L.; Broackes-Carter, F.; Campbell, N.H.; Chavali, G.; Chen, C.; del-Toro, N.; et al. The MIntAct project--IntAct as a common curation platform for 11 molecular interaction databases. Nucleic Acids Res. 2014, 42, D358-D363. doi: 10.1093/nar/gkt1115.
- Are figures 1CD, 2CD, and 3CD boxplots? I suggest the author add figure legends explaining what the box and whisker represent in the plots.
Answer:
Yes, we agree! According to referee #2 remark, we added the following sentence to the figure captions:
“The boxes represent the average entropy per cancer type and the whiskers correspond to their standard deviations.”
- DESeq2 and edgeR are the most widely used DEGs pipelines in this area; the author should benchmark these two tools.
Answer
Yes, we agree! This observation is redundant with an observation of referee #1; we now added the following contents to the manuscript:
To attend this observation, we divided the manuscript in three steps:
Step 1 is populated with the results of the previous version as there was no remark about them.
Step 2 is populated by comparison of RKM with DESeq2, and edger as well as TPM, UQ, Med, CPM, RLE, QN, TMM in combination with log2 fold change.
Step 3, we considered the results of Step 2 as a calibration step or a training step. Based on the best performing methodology identified in Step 2, we tested additional paired samples (BCLA, LUAD, COAD, and UCS) within the correlation between tumoral entropy and 5-year OS. This approach is presented at the end of the introduction with the following paragraph between lines 142 and 168 in the introduction:
“We validated this process in three steps: (i)Accordingly, across 8 types of cancers (475 patients) spanning 5-year OS rates from 30 % to 98%, We evaluated the performance of RPKM and Median normalization on a gene-by-gene basis or and by referencing the population of DEGs across 8 types of cancers (475 patients) spanning 5-year OS rates from 30 % to 98% according to previous studies (Conforte et al., 2019; Pires et al., 2021). (ii) We compared RPKM to seven read count normalizations, i.e., Transcript Per Million (TPM, Li and Dewey, 2011), Counts Per Million (CPM, Dillies et al., 2013), Median (Med, Dillies et al., 2013), Upper Quantile (UQ, Bullard et al. 2010), Relative log expression (RLE, Love et al., 2014), Quantile normalization (QN, Bolstad et al., 2003), and Trimmed mean of M-values (TMM, Robinson and Oshlack, 2010) as well as to two cross-sample distribution based methods, i.e., DESeq2 (Love et al. 2014) and edgeR (Robinson et al. 2010). (iii) Based on the best performing methodologies identified in these comparison, we tested paired samples from additional cancer types from TCGA to validate the approach, including bladder carcinoma (BLCA), lung adenocarcinoma (LUAD), colon adenocarcinoma (COAD), and uterine carcinoma (UCS).
According to this Bayesian learning process, we found the following: (i) We used tThe coefficient of correlation between average entropy per cancer type and aggressiveness (5-year OS) ais a suitable metric for the comparative performance of biological information extraction of sub-networks of up-regulated malignant genes. In our hands, (ii) TMM, QN, and RLE, a group of methods that determine a scaling factor for variation stabilization, produced a correlation coefficient similar to TPM, but with a standard deviation approximately 25% lower. The straightforward approach of combining a the RPKM normalization method (even without a scaling factor for variation stabilization) with log2 fold change yielded better average correlation coefficients than probabilistic methods such as DESeq2 and edgeR for determining network entropy. associated to the log2 fold change was the process that gave the most consistent results in terms of entropy.(iii) The correlation coefficient decreased from r ≈ 0.9 to r ≈ 0.6 when the number of cancer types increased from 8 to 12.”
Conforte AJ, Tuszynski JA, da Silva FAB, Carels N (2019) Signaling complexity measured by Shannon entropy and its application in personalized medicine. Front Genet 10, 930.
Pires JG, da Silva GF, Weyssow T, Conforte AJ, Pagnoncelli D, da Silva FAB, Carels N (2021) Galaxy and MEAN Stack to create a user-friendly workflow for the rational optimization of cancer chemotherapy. Front Genet 12, 624259.
Li, B.; Dewey, C.N. RSEM: accurate transcript quantification from RNA-Seq data with or without a reference genome. BMC Bioinformatics 2011, 12, 323. doi: 10.1186/1471-2105-12-323.
Dillies, M.-A.; Rau, A.; Aubert J.; Hennequet-Antier, C.; Jeanmougin, M.; Servant, N.; Keime, C.; Marot G.; Castel D.; Estelle, J.; et al. A comprehensive evaluation of normalization methods for Illumina high-throughput RNA sequencing data analysis. Briefings Bioinf. 2013, 14, 671–683. doi: 10.1093/bib/bbs046.
Bullard, J.H.; Purdom, E.; Hansen, K.D.; Dudoit, S. Evaluation of statistical methods for normalization and differential expression in mRNA-Seq experiments. BMC Bioinf. 2010, 11, 94. doi: 10.1186/1471-2105-11-94.
Bolstad, B.M.; Irizarry, R.A.; Åstrand, M.; Speed, T.P. A comparison of normalization methods for high density oligonucleotide array data based on variance and bias. Bioinformatics 2003, 19, 185–193. doi: 10.1093/bioinformatics/19.2.185.
Robinson, M.D.; Oshlack, A. A scaling normalization method for differential expression analysis of RNA-seq data. Genome Biol. 2010, 11, R25. doi: 10.1186/gb-2010-11-3-r25.
Robinson, M.D.; McCarthy, D.J.; & Smyth, G.K. edgeR: a Bioconductor package for differential expression analysis of digital gene expression data. Bioinformatics 2010, 26, 139-140. doi: 10.1093/bioinformatics/btp616.
In the Materials and Method section, we added:
“TPM: The gene expression in transcripts per million (TPM) of a gene is defined as the ratio of it RPK over the sum of all RPKs (per million) [38]. We computed raw counts in accordance with formula 2 using a custom Perl script derived from the one used for RPKM calculation.
2.4. Extended normalization methods
In this section, we aim to compare the normalization methods provided by the NormSeq server (https://arn.ugr.es/normSeq) to RPKM and TPM. Briefly and citing Scheepbouwer et al. (2023), the purpose of these methods can be summarized as follow:
Counts Per Million (CPM): CPM normalization corrects for library size without considering transcript length (Dillies et al., 2013).
Median (Med): Median normalization adjusts the data of each individual sample by adding a constant value to achieve the same median value across all samples (Dillies et al., 2013).
Upper Quantile (UQ): All genes with a read count of 0 are removed, followed by a division of the remaining gene counts by the upper quartile (Bullard et al. 2010).
Relative log expression (RLE): For each gene, the RLE scaling factor is computed as the median of the ratio of the read counts by taking the geometric mean across all samples (Love et al., 2014).
Quantile normalization (QN): Quantile normalization applies a mathematical transformation to the rank statistics across samples (Bolstad et al., 2003).
Trimmed mean of M-values (TMM): The TMM method estimates scale factors for comparing libraries on a relative scale (Robinson and Oshlack, 2010).
2.5. Differential expression method
Here we considered DESeq2 (Love et al. 2014) and edgeR (Robinson et al. 2010) as reference software for benchmarking the capacity of entropy to report on the extraction of biological information given the complexity of sub-networks associated with malignant up-regulated genes. Both packages are cross-sample distribution-based methods that estimate the dispersion parameter for each gene, reflecting the variability of read counts according to the negative binomial distribution. These software apply the Benjamini-Hochberg procedure to control the false discovery rate (FDR), helping manage the multiple testing problem inherent in RNA-seq data analysis. However, they differ in their normalization methods: DESeq2 is based on the median of ratios to normalize read counts (MRN) and EdgeR is based on trimmed mean of M-values (TMM). DESeq2 was run from the iDEP server (http://bioinformatics.sdstate.edu/idep/) (Ge et al., 2018) and edgeR from NormSeq (https://arn.ugr.es/normSeq).
”
In the Results section, we added the following data:
“3.2. Step 2: Comparison of normalization methods and differential expression determination processes
The results in Table 2 were produced by replacing RPKM by TPM in the pipeline of Figure 1A and replacing the normalization method step in Figure 1B with UQ, Med, CPM, RLE, QN, or TMM (Table S3). Table 2 shows that the correlation coefficient between entropy and 5-year OS for the sub-networks of malignant up-regulated genes of eight cancer types was lower for DESeq2 and edgeR (r = 0.72) compared to the correlation obtained with log2 fold change ≥ 1 filter applied to raw count normalized with any methods (even with no normalization). RLE and PCA plots are given in Figure S2 for DESeq2.
Table 2. Comparison of RPKM to seven normalization methods and two differentially expressed methods (DEG).
|
|
|
|
|
|
|
Normalization methods |
||||||
|
|
|
DEG method |
NN1 |
|
RPKM |
|
TPM |
|
UQ |
|
||
|
Cancer |
5-y. OS |
DESeq2 |
edgeR |
Av. |
St. Dev. |
Av. |
St. Dev. |
Av. |
St. Dev. |
Av. |
St. Dev. |
|
|
STAD |
37.67 |
2.277 |
1.452 |
3.217 |
0.751 |
3.114 |
0.432 |
3.222 |
0.468 |
3.065 |
0.733 |
|
|
LUSC |
47.25 |
3.028 |
1.399 |
3.158 |
0.451 |
3.168 |
0.405 |
3.221 |
0.420 |
2.472 |
0.619 |
|
|
LIHC |
48.63 |
2.470 |
1.370 |
3.452 |
0.719 |
3.460 |
0.551 |
3.351 |
0.551 |
3.178 |
0.862 |
|
|
KIRC |
63.24 |
2.482 |
1.224 |
2.666 |
0.545 |
2.671 |
0.293 |
2.659 |
0.318 |
2.083 |
0.624 |
|
|
KIRP |
75.28 |
2.406 |
1.449 |
2.654 |
0.612 |
2.594 |
0.408 |
2.613 |
0.452 |
2.083 |
0.611 |
|
|
BRCA |
81.90 |
2.362 |
1.374 |
2.812 |
0.600 |
2.720 |
0.468 |
2.731 |
0.493 |
2.377 |
0.655 |
|
|
THCA |
93.02 |
1.905 |
1.084 |
2.354 |
0.577 |
2.166 |
0.340 |
2.133 |
0.360 |
2.002 |
0.711 |
|
|
PRAD |
97.83 |
1.445 |
0.738 |
2.172 |
0.764 |
2.097 |
0.526 |
2.073 |
0.534 |
1.945 |
0.720 |
|
|
|
|
|
|
|
|
|
|
|
|
|
|
|
|
Correl. |
|
-0.722 |
-0.720 |
-0.910 |
|
-0.914 |
|
-0.942 |
|
-0.814 |
|
|
|
Av. |
|
|
|
|
0.627 |
|
0.428 |
|
0.449 |
|
0.692 |
|
|
St. Dev. |
|
|
|
|
0.109 |
|
0.087 |
|
0.081 |
|
0.084 |
|
1NN stands for “no normalization”.
Table 2 (continued). Comparison of RPKM to seven normalization methods and two DEG methods.
|
|
|
Normalization methods |
|||||||||
|
|
|
Med |
|
CPM |
|
RLE |
|
QN |
|
TMM |
|
|
Cancer |
5-y. OS |
Av. |
St. Dev. |
Av. |
St. Dev. |
Av. |
St. Dev. |
Av. |
St. Dev. |
Av. |
St. Dev. |
|
STAD |
37.67 |
3.100 |
0.764 |
3.105 |
0.433 |
3.110 |
0.307 |
3.217 |
0.257 |
3.108 |
0.319 |
|
LUSC |
47.25 |
3.181 |
0.449 |
2.487 |
0.398 |
3.161 |
0.354 |
3.100 |
0.351 |
3.162 |
0.361 |
|
LIHC |
48.63 |
3.399 |
0.733 |
3.439 |
0.558 |
3.495 |
0.175 |
3.560 |
0.208 |
3.497 |
0.202 |
|
KIRC |
63.24 |
2.690 |
0.547 |
2.663 |
0.291 |
2.652 |
0.255 |
2.641 |
0.203 |
2.654 |
0.244 |
|
KIRP |
75.28 |
2.654 |
0.612 |
2.588 |
0.408 |
2.587 |
0.270 |
2.633 |
0.259 |
2.588 |
0.408 |
|
BRCA |
81.90 |
2.682 |
0.599 |
2.693 |
0.472 |
2.690 |
0.307 |
2.757 |
0.307 |
2.686 |
0.309 |
|
THCA |
93.02 |
2.214 |
0.561 |
2.152 |
0.338 |
2.154 |
0.311 |
2.285 |
0.259 |
2.157 |
0.325 |
|
PRAD |
97.83 |
2.046 |
0.770 |
2.070 |
0.528 |
2.095 |
0.380 |
2.150 |
0.406 |
2.098 |
0.379 |
|
|
|
|
|
|
|
|
|
|
|
|
|
|
Correl. |
|
-0.926 |
|
-0.777 |
|
-0.911 |
|
-0.898 |
|
-0.911 |
|
|
Av. |
|
|
0.629 |
|
0.428 |
|
0.295 |
|
0.281 |
|
0.318 |
|
St. Dev. |
|
|
0.116 |
|
0.090 |
|
0.063 |
|
0.070 |
|
0.068 |
This table also shows that TPM (r = 0.94) ranks highest among methods without variance stabilization by a scaling factor, such as RPKM (r = 0.91), UQ (r = 0.81), Med (r = 93), and CPM (r = 0.77). Some of these methods performed even better than variance-stabilized methods (RLE: r = 0.91, QN: r = 0.90, TMM: r = 0.91), but they exhibited nearly double the rate of average standard deviation (0.4 to 0.6 compared to ~0.3).
The variance stabilization of RLE, QN, and TMM normalizations can be verified from their RLE plots (Gandolfo, 2018) compared to TPM, RPKM, UQ, Med, and CPM (Figures S3). However, variance stabilization does not necessarily improved the PCA classification of control and tumor samples (Figures S4); in some cases PCA classification is effective, and in other, it is not, without any apparent correlation to any specific feature.
Gandolfo, L.C.; Speed, T.P. RLE plots: Visualizing unwanted variation in high dimensional data. PLoS One 2018, 13, e0191629. doi: 10.1371/journal.pone.0191629.
“3.3. Step 3: Generalization of the degree-entropy vs. 5-year OS relationship
Since the eight cancer type in Table 2 could be the result of an over-fitting process, we include paired samples of four additional cancer types: BLCA, LUAD, COAD, and UCS. Table 3 shows that it is indeed the case; however, the negative relationship between entropy and 5-year OS is maintained. The best result was obtained with TPM (r = –0.674), while RLE (r = –0.602) and TMMM (r = –0.598) exhibited similar lower correlation coefficients but also lower variance.
Table 3. Comparison of the best performing methods for RNA-seq of 12 cancer types.
|
|
|
Normalization methods |
|
|
|
|
|
|
|||
|
|
|
RPKM |
|
TPM |
|
Med |
|
RLE |
|
TMM |
|
|
Cancer |
5-y. OS |
Av. |
St. Dev. |
Av. |
St. Dev. |
Av. |
St. Dev. |
Av. |
St. Dev. |
Av. |
St. Dev. |
|
BLCA |
20.00 |
3.103 |
0.368 |
3.146 |
0.398 |
2.847 |
0.641 |
3.023 |
0.288 |
3.018 |
0.286 |
|
STAD |
37.67 |
3.114 |
0.432 |
3.222 |
0.468 |
3.100 |
0.764 |
3.110 |
0.307 |
3.108 |
0.319 |
|
LUAD |
40.00 |
2.500 |
0.452 |
2.499 |
0.479 |
2.297 |
0.785 |
2.355 |
0.427 |
2.355 |
0.429 |
|
LUSC |
47.25 |
3.168 |
0.405 |
3.221 |
0.420 |
3.181 |
0.449 |
3.161 |
0.354 |
3.162 |
0.361 |
|
LIHC |
48.63 |
3.460 |
0.551 |
3.351 |
0.551 |
3.399 |
0.733 |
3.495 |
0.175 |
3.497 |
0.202 |
|
KIRC |
63.24 |
2.671 |
0.293 |
2.659 |
0.318 |
2.690 |
0.547 |
2.652 |
0.255 |
2.654 |
0.244 |
|
COAD |
68.45 |
2.943 |
0.360 |
2.943 |
0.390 |
2.650 |
1.191 |
2.878 |
0.329 |
2.887 |
0.308 |
|
KIRP |
75.28 |
2.594 |
0.408 |
2.613 |
0.452 |
2.654 |
0.612 |
2.587 |
0.270 |
2.588 |
0.408 |
|
BRCA |
81.90 |
2.720 |
0.468 |
2.731 |
0.493 |
2.682 |
0.599 |
2.690 |
0.307 |
2.686 |
0.309 |
|
UCS |
89.27 |
2.948 |
0.285 |
2.946 |
0.332 |
3.323 |
0.471 |
2.866 |
0.197 |
2.868 |
0.192 |
|
THCA |
93.02 |
2.166 |
0.340 |
2.133 |
0.360 |
2.214 |
0.561 |
2.154 |
0.311 |
2.157 |
0.325 |
|
PRAD |
97.83 |
2.097 |
0.526 |
2.073 |
0.534 |
2.046 |
0.770 |
2.095 |
0.380 |
2.098 |
0.379 |
|
|
|
|
|
|
|
|
|
|
|
|
|
|
Correl. |
|
-0.643 |
|
-0.674 |
|
-0.397 |
|
-0.602 |
|
-0.598 |
|
|
Av. |
|
|
0.407 |
|
0.433 |
|
0.677 |
|
0.300 |
|
0.313 |
|
St. Dev. |
|
|
0.084 |
|
0.076 |
|
0.198 |
|
0.071 |
|
0.075 |
By plotting the relationship entropy vs 5-year OS for TPM (Figure 5A) and RLE (Figure 5B), one can better visualize the lower variation associated to RLE compared to TPM. Both relationships are very similar. The decrease in the correlation coefficient from TPM to RLE or TMM is due to the decrease in covariance. The covariance for TPM was –6.458 while for RLE, it was –5763. From this, one may conclude that variance stabilization through the application of a scaling factor has a negative effect on the correlation.
Fig. 5. Relationship of degree entropy of the sub-network formed by the up-regulated malignant genes vs their corresponding patients’ 5year OS. A. TPM (r = –0.67; y=–0.0114*x+3.518). B. RLE (r = –0.60; y=–0.0101*x+3.401). 1BLCA, 2STAD, 3LUAD, 4LUSC, 5LIHC, 6KIRC, 7COAD, 8KIRP, 9BRCA, 10UCS, 11THCA, 12PRAD. The boxes represent the average entropy per cancer type and the whiskers correspond to their standard deviations.”
In the Discuttion section, we added:
“4.3. Comparison of normalization methods and differential expression determination processes
Interestingly we found that DESeq2 and edgeR were not effective in assessing the complexity of biological network across a large cohort encompassing a 5-year OS aggressiveness rate ranging from 20 to 98%. From this finding, one might conclude that methods relying on negative binomial distribution and Benjamini-Hochberg procedure for false discovery rate (FDR) control are not optimal for extracting biological complexity from RNA-seq data. Conversely, normalization methods such as TPM, RLE, and TMM combined with log2 fold change appear suitable for this purpose.
4.4. The relationship of degree-entropy and 5-year OS
It is interesting to note that the negative correlation between degree-entropy and 5-year OS, albeit this correlation diminishes notably when analyzing paired samples of 12 cancer types rather than eight. The underperformance of Med in this context indicates its inadequate adaptation to biological samples characterized by significant topological complexity variation. This underscores the presence of over-fitting in the previous analyses. Nevertheless, the observed reduction in correlation remains intriguing as it suggests that tumors adjusted their gene up-regulation patterns to form more o less complex sub-networks depending on the cancer type and its environment. This consideration holds importance in guiding clinical decisions regarding optimal therapeutic strategies. For instance, therapies targeting hubs may be less effective in tumors exhibiting lower entropy levels.
Among the types of errors that could account for the correlation reduction across the 12 cancer types, it should be noted that the standard deviation associated with the x axis is unknown. Although 5-year OS is a statistical metric intended to ensure robustness of the data on the x axis, uncertainties remain. Another factor to consider is that the efficacy of treatments for specific cancer type can vary independently of their aggressiveness. This variability can influence the 5-year OS, which have generally increased over time but at differing rates depending on the type of cancer.
”
- The author mentioned that cancer-enriched PPI networks are more redundant in aggressive cancers. Therefore, the entropy-survival rates correlation benchmark methods can only be used when analyzing other cancer-related RNA-seq data, not all RNA-seq analyses. The author should clarify this in the paper.
Answer:
Yes, we agree! We would expect that if the workflow is optimized for cancer RNA-seq, it would also work for RNA-seq from other biological systems. Indeed, the only difference could be in the maping process because of huge number of mutations, fusion and indel events that may occur in genes affected by cancer. Thus, cancer being the worst case for biological information extraction by RNA-seq, we do not glimpse why a workflow optimized with tumor samples would not be suitable for RNA-seq aiming at describing other biological situations. Refering specifically to entropy as a measure of topological complexity for disease characterization let’s quote Hu et al. (2017): “the hubs registered significant enrichment (considering protein “participation”) for 11 out of the considered 18 classes of diseases. They include cancers and diseases of the stomatognathic, endocrine, digestive, respiratory, female urogenital, nervous, and musculoskeletal systems, each associating with at least 100 human proteins”.
Given these considerations, we added the following paragraph to the manuscript in the discussion:
“According to Hu et al. (2017), hubs registered significant enrichment to the PPI network of 18 classes of diseases including those of the stomatognathic, endocrine, digestive, respiratory, female urogenital, nervous, and musculoskeletal systems, in addition to cancers. Thus, a correlation between network topology and certain features of these diseases could potentially be observed. However, we would expect that a workflow optimized for cancer RNA-seq should also be suitable for RNA-seq from other biological systems. The primary difference lies in the read mapping process due to the high frequency of mutation, fusion and indel events in cancer-affected genes. Given that cancer represents the most challenging case for biological information extraction via RNA-seq, we see no reason why a workflow optimized with tumor samples would not be suitable for RNA-seq aimed at describing other biological contexts.”
Hu, G.; Wu, Z.; Uversky, V.N.; Kurgan, L. Functional analysis of human hub proteins and their interactors involved in the intrinsic disorder-enriched interactions. Int. J. Mol. Sci. 2017, 18, 2761. doi: 10.3390/ijms18122761.
Round 2
Reviewer 2 Report
Comments and Suggestions for Authors
The author addressed all the questions.